# Enhancing the Shelf Life of Sous-Vide Red Deer Meat with *Piper nigrum* Essential Oil: A Study on Antimicrobial Efficacy against *Listeria monocytogenes*

**DOI:** 10.3390/molecules29174179

**Published:** 2024-09-03

**Authors:** Miroslava Kačániová, Natália Čmiková, Zhaojun Ban, Stefania Garzoli, Joel Horacio Elizondo-Luevano, Anis Ben Hsouna, Rania Ben Saad, Alessandro Bianchi, Francesca Venturi, Maciej Ireneusz Kluz, Peter Haščík

**Affiliations:** 1Institute of Horticulture, Faculty of Horticulture and Landscape Engineering, Slovak University of Agriculture, Trieda Andreja Hlinku 2, 94976 Nitra, Slovakia; n.cmikova@gmail.com; 2School of Medical & Health Sciences, University of Economics and Human Sciences in Warsaw, Okopowa 59, 01043 Warszawa, Poland; m.kluz@vizja.pl; 3Zhejiang Provincial Key Laboratory of Chemical and Biological Processing Technology of Farm Products, Zhejiang Provincial Collaborative Innovation Center of Agricultural Biological Resources Biochemical Manufacturing, School of Biological and Chemical Engineering, Zhejiang University of Science and Technology, Hangzhou 310023, China; banzhaojun@zust.edu.cn; 4Department of Chemistry and Technologies of Drug, Sapienza University, P. le Aldo Moro, 5, 00185 Rome, Italy; stefania.garzoli@uniroma1.it; 5Faculty of Agronomy, Universidad Autónoma de Nuevo León (UANL), Av. Francisco Villa S/N, Col. Ex Hacienda el Canadá, General Escobedo, Nuevo León 66050, Mexico; joel.elizondolv@uanl.edu.mx; 6Laboratory of Biotechnology and Plant Improvement, Centre of Biotechnology of Sfax, B.P “1177”, Sfax 3018, Tunisia; benhsounanis@gmail.com (A.B.H.); raniabensaad@gmail.com (R.B.S.); 7Department of Environmental Sciences and Nutrition, Higher Institute of Applied Sciences and Technology of Mahdia, University of Monastir, Monastir 5000, Tunisia; 8Department of Agriculture, Food and Environment, University of Pisa, Via del Borghetto 80, 56124 Pisa, Italy; francesca.venturi@unipi.it; 9Institute of Food Technology, Faculty of Biotechnology and Food Sciences, Slovak University of Agriculture, Trieda Andreja Hlinku 2, 94976 Nitra, Slovakia; peter.hascik@uniag.sk

**Keywords:** black pepper essential oil, *musculus biceps femoris*, shelf life, storage, *Listeria monocytogenes*, sous-vide technique, vacuum packaging

## Abstract

Using sous-vide technology in combination with essential oils offers the potential to extend the preservation of food items while preserving their original quality. This method aligns with the growing consumer demand for safer and healthier food production practices. This study aimed to assess the suitability of minimal processing of game meat and the effectiveness of vacuum packaging in combination with *Piper nigrum* essential oil (PNEO) treatment to preserve red deer meat samples inoculated with *Listeria monocytogenes*. Microbial analyses, including total viable count (TVC) for 48 h at 30 °C, coliform bacteria (CB) for 24 h at 37 °C, and *L. monocytogenes* count for 24 h at 37 °C, were conducted. The cooking temperature of the sous-vide was from 50 to 65 °C and the cooking time from 5 to 20 min. Additionally, the study monitored the representation of microorganism species identified through mass spectrometry. The microbiological quality of red deer meat processed using the sous-vide method was monitored over 14 days of storage at 4 °C. The results indicated that the TVC, CB, and *L. monocytogenes* counts decreased with the temperature and processing time of the sous-vide method. The lowest counts of individual microorganism groups were observed in samples treated with 1% PNEO. The analysis revealed that PNEO, in combination with the sous-vide method, effectively reduced *L. monocytogenes* counts and extended the shelf life of red deer meat. *Kocuria salsicia*, *Pseudomonas taetrolens*, and *Pseudomonas fragi* were the most frequently isolated microorganism species during the 14-day period of red deer meat storage prepared using the sous-vide method.

## 1. Introduction

The market continues to witness the emergence of novel meat products, including those derived from horses, deer, rabbits, ostriches, wild yaks, and game animals [1,2,3,4]. Game meat often takes precedence over meat from domesticated animals [5]. This preference stems from the perception of game meat as a natural food. It is believed that wild animals graze as nature intended, possess greater strength due to natural selection, and experience less stress as they roam freely and follow their instincts [6]. *Cervus elaphus* meat exhibits variations in fat content ranging from 1.1 to 3.9% and boasts low cholesterol levels. It is also abundant in minerals, essential amino acids, vitamins, and protein [7]. Furthermore, red deer meat boasts a distinct array of bioactive compounds, ferments, and hormones that are thought to confer health benefits. While the mineral composition of red deer meat may share similarities with beef, specific elements are present in greater abundance in red deer meat compared to cattle meat. Red deer meat notably exceeds beef in its levels of calcium, fluorine, iron, copper, zinc, and chromium [8]. Moreover, red deer meat serves as an excellent source of vitamins A, B, C, and E, alongside minerals including iron (Fe), potassium (K), calcium (Ca), magnesium (Mg), copper (Cu), zinc (Zn), and selenium (Se) [9,10]. Bioactive elements such as hormones and ferments found in meat can prove beneficial for individuals in a weakened state. Approximately 55 to 60% of the animal’s weight comprises meat [11].

*Listeria monocytogenes* was selected as a model organism for thermal inactivation tests. *L. monocytogenes* is known for its high tolerance to pH, salt, and heat [12,13]. Moreover, the prevalence of listeriosis has been increasing in European nations, the highest mortality rate among zoonotic illnesses under surveillance (16.2%) [14]. This is particularly concerning for consumers, as *L. monocytogenes* is commonly found in wild game meat. Studies have reported contamination rates of 4.8% in newly shot game meat samples [14] and a comparatively lower prevalence of 2.8% in game meat [15]. In Eastern Europe, Paulsen et al. [16] found *L. monocytogenes* in 23.5% of wild boar meat and 12.5% of wild game meat samples. Also, they discovered that 9% of roe deer samples from a game processing establishment tested positive for *L. monocytogenes*.

The sous-vide (SV) method stands out among various food heat treatment techniques and is particularly renowned for its efficacy in vacuum-sealed environments, where foods undergo pasteurization to prolong their shelf life [17]. The intensity of heat treatment is closely associated with the exposure temperature, which is expected to affect bacteria, including spore-forming bacteria. The presence of microbes in food constitutes a major factor behind food product recalls, outbreaks of foodborne illnesses, and food spoilage. Consequently, it is imperative to employ efficient preservatives and preservation techniques to control bacterial growth and mitigate their effects on food quality [18]. It is essential to thoroughly analyze and evaluate the SV technique’s ability to eradicate spore-forming microorganisms and spores, enhance food safety, and extend shelf life. Certain critics have expressed apprehensions regarding the relatively low temperature range employed in SV technology, suggesting that it might not be adequate to guarantee the microbiological safety of food when utilized as a preservation method [19]. Nevertheless, research into the advantages of the technique in prolonging shelf life has demonstrated that SV treatment effectively decreases microbial growth [20]. Thus, a synergistic effect has been proposed; for instance, integrating SV with other preservation methods like essential oils may offer enhanced effectiveness in extending the shelf life of food items [21]. Combining stabilization with essential oils has been observed to effectively inhibit the proliferation of *L. monocytogenes*. Employing SV in conjunction with essential oils resulted in the deactivation of contaminating bacteria in beef meat [22].

Chemical food preservatives have traditionally been used to effectively manage spoilage and harmful germs. However, finding nutritious foods without artificial preservatives can be difficult. To meet consumer preferences, the food industry has turned its attention to natural antibacterial compounds like essential oils [23]. Today’s consumers expect non-toxic and natural products, which require the protection of food from germs during storage [24]. Plants contain bioactive chemicals such as phenolic compounds, alkaloids, flavonoids, steroids, and terpenes, many of which have been studied for their antibacterial and biological properties [25]. Pepper extracts and volatile oils have been found to help prevent food spoilage and infections according to numerous studies [26,27,28]. For instance, raw pepper possesses phenolic compounds capable of hindering the proliferation of *Staphylococcus aureus*, *Salmonella typhimurium*, *Escherichia coli* and various *Bacillus* species [29]. Pepper, scientifically known as *Piper nigrum* L., was first encountered in China and originated in India [30]. Besides its use as a spice, pepper has medicinal properties. Pepper essential oil contains alkaloids, terpenes, flavones, and volatile oils like piperine, each with unique biological activities [31,32,33]. Thiel et al. [28] noted that pepper is also used to enhance flavor and preserve meat and meat-based products.

The research focused on examining the efficiency of vacuum sealing in conjunction with *Piper nigrum* essential oil (PNEO) treatment to preserve red deer meat samples for sous-vide cooking and to assess the suitability of minimal meat processing. The effectiveness of these methods was evaluated by analyzing quality indicators and quantifying *Listeria monocytogenes* in red deer meat samples inoculated with the bacteria during refrigerated storage.

## 2. Results

### 2.1. Microbial Counts

The total viable count (TVC) of red deer meat samples subjected to various temperature, time, PNEO, and *L. monocytogenes* treatments is presented in Appendix A and Table 1. Control samples comprised raw, uncooked, and unpacked red deer meat, with assessments conducted on day 0, yielding a TVC of 3.25 log CFU/g and zero coliform bacteria. In the control group of sous-vide red deer meat samples, the TVC varied on day 1 from 2.18 log CFU/g for the group treated at 60 °C for 5 min to 3.52 log CFU/g for the group treated at 50 °C for 5 min. On day 7, the TVC ranged from 1.22 log CFU/g for the group treated at 65 °C for 5 min to 3.79 log CFU/g for the group treated at 50 °C for 5 min; on day 14, it ranged from 1.20 log CFU/g for the group treated at 65 °C for 20 min to 3.88 log CFU/g for the group treated at 50 °C for 5 min. In the vacuum-packaged sous-vide deer samples control group, the TVC was lower compared to the non-vacuum-packaged control group (Table 1). On day 1, it ranged from 1.86 log CFU/g for the group treated at 55 °C for 20 min to 3.43 log CFU/g for the group treated at 50 °C for 5 min; on day 7, it ranged from 1.87 log CFU/g for the group treated at 60 °C for 5 min to 3.54 log CFU/g for the group treated at 50 °C for 5 min; on day 14, it ranged from 1.16 log CFU/g for the group treated at 65 °C for 5 min to 3.71 log CFU/g for the group treated at 50 °C for 5 min. Lower TVCs were observed in the PNEO-treated groups, ranging on day 1 from 1.84 log CFU/g for the group treated at 55 °C for 20 min to 3.20 log CFU/g for the group treated at 50 °C for 5 min (Table 1); on day 7, they ranged from 1.77 log CFU/g for the group treated at 60 °C for 5 min to 3.40 log CFU/g for the group treated at 50 °C for 5 min; and, on day 14, they ranging from 1.38 log CFU/g for the group treated at 60 °C for 20 min to 3.60 log CFU/g for the group treated at 50 °C for 5 min. In sous-vide red deer meat samples inoculated with *L. monocytogenes*, the TVC ranged on day 1 from 1.7 log CFU/g for samples treated at 60 °C for 5 min to 3.45 log CFU/g for samples treated at 50 °C for 5 min; on day 7, it ranged from 1.96 log CFU/g for samples treated at 60 °C for 5 min to 3.77 log CFU/g for samples treated at 50 °C for 5 min (Table 1). The last group, with *L. monocytogenes* inoculation and PNEO treatment, exhibited TVCs ranging on day 1 from 1.86 log CFU/g for samples treated at 55 °C for 20 min to 3.42 log CFU/g for samples treated at 50 °C for 5 min. On day 7, the TVC ranged from 1.97 log CFU/g for samples treated at 55 °C for 20 min to 3.66 log CFU/g for samples treated at 50 °C for 5 min, and, on day 14, the TVC ranged from 1.03 for samples treated at 65 °C for 20 min to 3.80 log CFU/g for samples treated at 50 °C for 5 min.

Appendix A illustrates the count of coliform bacteria (CB) in sous-vide red deer meat samples. On the first day, the CB counts in all treated groups were zero, similar to day 0. CB were only detected in deer sous-vide samples on days 7 and 14. On day 7, in the control group, the CB counts ranged from 2.25 log CFU/g for the group treated at 55 °C for 10 min to 3.61 log CFU/g for the group treated at 50 °C for 5 min. In the control group with vacuum packaging, CB counts ranged from 1.33 log CFU/g for the group treated at 55 °C for 5 min to 2.87 log CFU/g for the group treated at 50 °C for 5 min (Table 2). Sous-vide red deer meat samples treated with PNEO exhibited lower CB counts than previous control groups. On day 7, the CB counts ranged from 1.22 log CFU/g to 2.34 log CFU/g in the first groups treated at 50 °C for 5 and 10 min. The group inoculated with *L. monocytogenes* showed CB counts ranging from 1.87 log CFU/g to 2.21 log CFU/g in the groups treated at 50 °C for 5 and 10 min, while in the group inoculated with *L. monocytogenes* in combination with PNEO, the CB counts were 1.86 log CFU/g in the group at 50 °C for 5 min. On day 14, the CB counts in the control group without vacuum packaging ranged from 2.60 log CFU/g for the group treated at 55 °C for 10 min to 3.77 log CFU/g for the group treated at 55 °C for 10 min. In the control group with vacuum packaging, the CB counts varied from 1.58 log CFU/g for the group treated at 55 °C for 5 min to 3.08 log CFU/g for the group treated at 50 °C for 5 min. In the group inoculated with PNEO treatment, CB counts ranged from 1.59 log CFU/g for the group at 50 °C for 15 min to 2.55 log CFU/g for the group treated at 50 °C for 5 min (Table 2). The CB counts in the group with sous-vide red deer meat samples inoculated with *L. monocytogenes* ranged from 1.87 log CFU/g (50 °C for 20 min) to 3.12 log CFU/g (50 °C for 5 min), while, in the group with sous-vide red deer meat samples inoculated with *L. monocytogenes* and PNEO, the CB counts ranged from 1.77 log CFU/g (50 °C for 15 min) to 2.31 log CFU/g (50 °C for 5 min). In general, there was no significant difference (*p* > 0.05) in the CB count on day 1; however, on days 7 and 14 post-incubation, they behaved similarly. It was determined that there was no significant difference in the CB count in the 50 and 55 °C treatments; however, a high significance (*p* ≤ 0.05) was observed in the 60 and 65 °C treatments (Table 2).

Table 3 shows that there is a significant difference (*p* ≤ 0.05) between the treatments at 60 and 65 °C compared to the treatments evaluated at 50 and 55 °C in the *L. monocytogenes* counts determined at 1, 7, and 14 days later. The count of *L. monocytogenes* was detected only in the last two groups on all storage days, as shown in Appendix A. In the group inoculated with *L. monocytogenes*, on day 1, the counts ranged between 2.20 log CFU/g (50 °C for 15 min) and 3.17 log CFU/g (50 °C for 5 min), while, in the group with samples inoculated with *L. monocytogenes* and treated with PNEO, the counts ranged between 2.32 log CFU/g (50 °C for 10 min) and 2.89 log CFU/g (50 °C for 5 min) (Table 3). On day 7, the counts varied between 1.28 log CFU/g (55 °C for 5 min) and 2.83 log CFU/g (50 °C for 5 min) in the group with *L. monocytogenes*, and, in the group with *L. monocytogenes* treated with PNEO, the counts ranged between 1.97 log CFU/g (50 °C for 15 min) and 2.56 log CFU/g (50 °C for 5 min). On day 14, the count of *L. monocytogenes* in the group inoculated with bacteria ranged between 1.36 log CFU/g (55 °C for 5 min) and 3.10 log CFU/g (50 °C for 5 min), while in the group inoculated with bacteria and treated with PNEO, the counts ranged between 2.09 log CFU/g (50 °C for 15 min) and 2.69 log CFU/g (50 °C for 5 min) (Table 3).

### 2.2. Microbial Strains Isolated for Red Deer Meat Samples

The TVC was only found in samples on day 0 (Figure 1). On this day, 36 isolates of microorganisms, including nine species, seven genera, and seven species, were identified in raw red meat deer. The most isolated species were *Acinetobacter guillouiae* (28%), followed by *Pseudomonas fragi* (14%), *Hafnia alvei*, and *Pantoea agglomerans* (11%).

On day 1, across all evaluated groups of organisms, 156 isolates with scores greater than 2.000 were identified. A total of 21 species, 12 genera, and 12 families was identified in the identified isolates (Figure 2). The most isolated species on day 1 was *L. monocytogenes* (28%) from the groups inoculated with these bacteria. Regardless, these bacteria were the most isolated species *P. fragi* (13%), followed *K. salsicia* (12%), *L. ivanovii* (10%), *P. lundensis*, and *P. taetrolens* (6%).

On the seventh day, 383 isolates with scores of more than 2,.000 were found across all assessed classes of microorganisms. There were 23 species, 12 genera, and 11 families found in all of the isolated isolates (Figure 3). Among the groups injected with these bacteria, *L. monocytogenes* accounted for 14% of the most isolated species on day 7. In any case, *P. taetrolens* and *K. salsicia* accounted for 12% of the most isolated species of bacteria, with *P. fragi* (10%) and *P. lundensis* (9%) following closely behind.

On day 14, 384 isolates with high scores were discovered from all assessed groups of microorganisms. There were 19 species, 11 genera, and nine families found in all of the isolated isolates (Figure 4). Among the groups injected with these bacteria, *L. monocytogenes* accounted for 15% of the most isolated species on day 14. In any case, *P. fragi* accounted for 10% of the most isolated species of bacteria, with *S. liquefaciens* (9%), *P. lundensis* (9%), *C. braakii*, and *Hafnia alvei* (7%) following closely behind.

## 3. Discussion

Commercial fresh meat products can have their shelf life extended by utilizing sous-vide treatment and vacuum packaging in combination with PNEO. Additionally, this approach ensures adherence to hygienic practices during storage and distribution. However, limited knowledge exists regarding how vacuum packing affects the shelf life of red deer meat. Application of untested conditions or increased heat resistance in bacteria may lead to the survival of foodborne microorganisms post-cooking. Therefore, the aim of this research is to evaluate the safety of sous-vide cooking of *L. monocytogenes*-contaminated venison using black pepper essential oil. In addition, the study investigates the effect of the vacuum cooking procedure and subsequent refrigeration on pathogen viability over a 14-day period. The outcomes of this study hold significant implications for stakeholders within the burgeoning sous-vide cooking industry, facilitating informed decision-making and furnishing safe cooking guidelines for both consumers and retailers. In our experiment, we employed varied temperatures and cooking durations for the meat to analyze its microbiological quality. Temperature is a pivotal factor in diminishing the presence of harmful pathogens in food items. Nonetheless, factors like the thickness and size of the product influence the duration needed for it to attain the correct internal temperature [34].

The control samples consisted of raw, unprocessed, and unpackaged red deer meat, which were evaluated on day 0. The results indicated a total viable count (TVC) of 3.25 log CFU/g, with no coliform bacteria (CB) being detected. However, a separate study reported a lower TVC in red deer meat [35]. In our experiment, the number of deer samples in the vacuum-packed sous-vide control group was lower than that in the control group without vacuum packing on day 1. Microbiota diversity is influenced by various factors, including packing type, storage temperature and duration, and the extent of bacterial contamination. Research has demonstrated that vacuum packaging can prolong the shelf life of commercial fresh beef products [36]. However, it is important to fully understand the microbiological implications of this technique with respect to the control of pathogenic organisms in foods and the elongation of the shelf life of foods. Research on the shelf life extension effects [37] of the technique has demonstrated that microbial growth is reduced after treatment. Hence, a synergistic effect has been proposed; for instance, SV combined with other preservation treatments, such as high-pressure processing (HPP) for meat, could prove more effective for extending the shelf life of food products [38].

Throughout the storage period, the TVC increased in alignment with the temperature applied. In samples containing pathogenic bacteria *L. monocytogenes*, the numbers gradually escalated, yet they stabilized in comparison to the control group. Conversely, in groups infused with PNEO, the numbers declined over the storage period. However, the quantity of CB remained constant. Our investigation revealed that the highest concentration of CB was observed in both control groups, and the bacteria did not proliferate during storage in red deer meat cooked sous-vide. Various studies have concluded that vacuum-packed white-tailed red deer meat should not be stored at 4 °C for more than 14 days. This recommendation is based on the observed differences in microbial contamination levels among different carcasses [39]. In our investigation, the mean values of the TVC and Enterobacteriaceae did not surpass the reference levels established by Klein and Schütze [40]. According to several studies [14,16,41,42], the microbial loads in wild boar and roe deer meat are comparable to or higher than those in livestock animals [43,44]. Additionally, according to Johansson et al. [45], the spoilage of fresh meat is an undesirable process involving both chemical and biological interactions. Microbiota diversity is influenced by various factors, including packing type, storage temperature and duration, and the degree of bacterial contamination. Research has demonstrated that vacuum packaging extends the shelf life of commercial fresh beef products [36]. Additionally, it maintains product sanitation during distribution and storage by preventing the growth of aerobic microorganisms. Vacuum-packed, refrigerated-stored fresh meat typically contains facultatively anaerobic and psychrotrophic anaerobic bacteria, with lactic acid bacteria (LAB) frequently being predominant [46,47].

Although interhost transmission cannot be completely discounted, the identification of identical *L. monocytogenes* strains in deer points towards contamination originating from a shared food or environmental source. Occasional reports of *L. monocytogenes* in game animals and on game carcasses have been documented [41,48,49,50,51]. Our analysis findings revealed that *L. monocytogenes* in sous-vide game meat samples survived in the group that received only the pathogen at 50 °C for ten minutes throughout the seven days of storage. *L. monocytogenes* was able to withstand temperatures up to 55 °C for seven days of storage. However, the PNEO-treated group survived up to only 50 °C. A similar trend was observed when sous-vide red deer meat samples were stored for 14 days. In their research, Abel et al. [52] investigated the impact of sous-vide cooking temperatures ranging from 50 °C to 60 °C on the rate of *L. monocytogenes* inactivation. They introduced three strains of *L. monocytogenes* into nutrient broth and minced game meat from *Capreolus capreolus* and *Sus scrofa*, followed by sous-vide cooking at 50 °C, 55 °C, or 60 °C for varying durations. Their findings highlighted the significant influence of the surrounding matrix on the decimal reduction values (D-values). For the brain–heart infusion (BHI), the D-values were 125.5 min at 50 °C, 29.7 min at 55 °C, and 5.1 min at 60 °C. Meanwhile, the D-values for roe deer were 49.2 min, 14.9 min, and 3.7 min at the respective temperatures, and for wild boar they were 100.2 min, 23.8 min, and 4.2 min. Prior investigations have explored the effects of low-temperature cooking conditions on the inactivation of *L. monocytogenes* in beef and pig matrices. Some studies have indicated that the composition of the surrounding matrix influences *L. monocytogenes* inactivation. However, there is limited understanding regarding how the matrices of different game meat species impact *L. monocytogenes* inactivation at low temperatures [53,54].

The inhibitory effect of EOs on *L. monocytogenes* has been documented in numerous studies. The obtained results indicated that LM populations increased during seven and fourteen days of storage at 4 °C in the control groups but decreased when exposed to EO treatment. At concentrations of 0.5% and 1%, EOs limited the growth of LM in meat at both temperatures, with better effects at the higher dose of 1%. In conclusion, EOs slowed the growth rates of *L. monocytogenes* populations compared to control during 14 days of storage at 4 °C. Further data show the efficacy of EO (1% *v*/*w*) in a meat model against two levels of an *L. monocytogenes* cocktail (3 and 6 log CFU/g) combined with storage at 4 °C for 14 days [55].

The appropriate concentration of PNEO was proposed based on other knowledge from other types of meat as well as our findings. The optimal concentration for the application of EOs is from 0.5 to 2%. It was demonstrated that PNEO exhibited antibacterial properties against *L. monocytogenes*, *S. typhimurium*, and *P. aeruginosa*, which aligns with the findings of Nikolić et al. [56]. However, higher concentrations of PNEO were required in our study to inhibit bacterial growth. This observed variation may be attributed to the characteristics of the tested microorganisms and the chemical composition of the PNEO utilized [57]. Dhifi et al. [57] suggested that the presence of key components might contribute to the activity of PNEO; however, the antibacterial efficacy of the oil is likely not solely attributable to its primary ingredients. Our results underscore the natural antibacterial properties of PNEO.

Black pepper (*Piper nigrum* L.) is one of the main flavoring agents in meat processing [58]. With a relatively high percentage of terpenoids (limonene, α- and β-pinene and caryophyllene), EOs isolated from black pepper (BPEOs), show a strong antioxidant effect as well as a preservative effect against a broad spectrum of microorganisms. BPEOs were added as a natural preservative in fresh pork loin at concentrations of 0 to 0.5%. All batches were stored at 4 °C for 9 days [58]. The study showed that the BPEOs delayed lipid oxidation and reduced the growth of *Enterobacteriaceae* and *Pseudomonas* spp. in fresh pork. In another study, the effect of a BPEO coating (0.05 and 0.1%) on the lipid oxidation and sensory quality (aroma) of ham was examined. The authors suggested that the use of BPEOs has a strong potential to suppress lipid oxidation and improve the sensory acceptability of ham during long-term storage (4 months at room temperature). Overall, the results suggest that BPEOs could be used as natural antioxidants in meat products [59].

In our study, various species of bacteria were isolated from sous-vide game meat samples on different days using mass spectrometry. On day 0, the most isolated species was *Acinetobacter guillouiae*. Excluding *L. monocytogenes*, which was deliberately applied to deer meat carcasses, the most isolated species on the first day of storage was *K. salsicia*; on the seventh day it was *K. salsicia* and *P. taetrolens*, and on the 14th day it was *P. fragi*. Limited information exists regarding the microbiota present in game meat. Peruzy et al. [60] observed that the predominant families isolated from wild boar meat include *Pseudomonas* (77%), *Pantoea* (73%), *Escherichia* (59%), and *Acinetobacter* (55%), with a notable prevalence of *Salmonella* (32%). Asakura et al. [61] identified *Escherichia coli* serotypes producing Shiga toxin in venison meat samples, along with coliform *Escherichia coli* and bacteria from the genera *Acinetobacter* and *Arthrobacter* in wild boar meat. Zgomba Maksimovic et al. [62] found coliform bacteria and a significant presence of *Bacillus cereus* bacteria in deer sausage samples. While there are some overlapping species between the game meat samples analyzed in the study by Kunová et al. [35] and our study, such as *Bacillus cereus* and *Pseudomonas*, the profiles differ markedly and do not include coliform bacteria, indicating a higher level of meat hygiene. Based on a microbiological analysis of raw meat, certain game samples were found to contain *Y. enterocolitica* and *L. monocytogenes*, posing potential food-borne disease risks. It is important to note that a limited number of samples were analyzed, making it challenging to draw broad conclusions about microbial loads in game meat [14,15,39,41,63]. However, findings from Pires et al. [64] have indicated that game meat might harbor zoonotic agents. Hygiene standards such as TVC or Enterobacteriaceae are not feasible for hunted wildlife due to the absence of controlled slaughter and evisceration conditions, unlike farm animals such as cattle or pigs. Several studies [14,16,41,42] have suggested that microbial loads in wild boar and roe deer meat are comparable to or even higher than those in livestock animals [43,44].

## 4. Materials and Methods

### 4.1. Preparation of Samples of Red Deer Meat

This study used samples of deer (*Cervus elaphus*) meat from the *musculus biceps femoris*. The analysis of the thigh of a 5-year-old deer from Slovak hunting grounds revealed the following physical and chemical properties: water 72.86, fat 0.76, protein 22.35, cholesterol 0.038, all expressed in g/100 g, pH 5.68, A_w_ 0.915. A total of 4 kg of thigh flesh were collected and stored in a refrigerator before being transferred to the microbiological laboratory. The red deer meat was sliced into 5 g portions using a sterile knife, resulting in 723 samples. The samples were distributed across different dates as follows: three raw red deer meat samples on day 0; 240 control and treated red deer meat samples on day 1, day 7, and day 14 each. Each 5 g portion of red deer meat was divided into control and treatment groups, then vacuum-wrapped after being mixed with a 100 µL of 1% (*v*/*w*) solution of PNEO dissolved in sunflower oil. The vacuum packing process was conducted using a Concept vacuum packer from Choceň, Czech Republic. Control samples were packed in polyethylene bags, while a second control group was vacuum-packed. *Listeria monocytogenes* and PNEO were added to the prepared samples (5 g), specifically 100 µL of *L. monocytogenes* and 100 µL of 1% (*v*/*w*) PNEO. Care was taken to prevent contamination during the mixing process, which lasted approximately one minute. Subsequently, the samples were vacuum-packed.

The following control and experimental groups were included in our trial:(i)Control: After being placed in polythene bags, without a vacuum, red deer meat samples were processed in a water bath at 50–65 °C for 5–20 min. The samples were then stored at 4 °C for 2 weeks.(ii)Control vacuum: After being vacuum-packed in polyethylene bags, red deer meat samples were processed in a water bath for 5–20 min at 50 to 65 °C. The samples were then stored at 4 °C for 2 weeks.(iii)Essential oil: After being treated with 1% PNEO and vacuum-packed, red deer meat samples were processed in a water bath for 5–20 min at 50 to 65 °C. The samples were then stored at 4 °C for 2 weeks.(iv)*Listeria monocytogenes*: After being inoculated with *L. monocytogenes* and vacuum-packed, red deer meat samples were processed in a water bath for 5–20 min at 50 to 65 °C. The samples were then stored at 4 °C for 2 weeks.(v)Essential oil + *Listeria monocytogenes*: After being treated with 1% PNEO, inoculated with *L. monocytogenes* and vacuum-packed red deer meat samples were processed in a water bath for 5–20 min at 50 to 65 °C. The samples were then stored at 4 °C for 2 weeks.

On day zero, control samples were prepared using raw, uncooked red deer meat. These samples were treated by carefully mixing and macerating them for 24 h with PNEO in one group and *L. monocytogenes* in another group. The samples were then processed using a CASO SV1000 sous-vide machine from Arnsberg, Germany. They were divided into groups and subjected to sous-vide treatment under carefully monitored temperature and time parameters. For packaging, vacuum-packed polyethylene high barrier bags were used. These bags were constructed from material ranging from 40 to 200 microns in size, providing impermeability, moisture-resistance, and high temperature resistance (−30 °C to +100 °C). They offered a long lifespan, maintaining the integrity of weld seams without softening, and were safe for storing food in the refrigerator for several years. Importantly, these bags were 100% free of plasticizers such as bisphenol A and did not contain microplastics, as indicated in the datasheet.

### 4.2. Bacteria Strain Preparation

The experiment utilized *Listeria monocytogenes* CCM 4699 obtained from the Czech Collection of Microorganisms in Brno, Czech Republic. The bacterial inoculum was cultured on Mueller–Hinton agar (MHA) from Oxoid in Basingstoke, UK, for 24 h at 37 °C. Following this, the inoculum’s optical density was adjusted to the 0.5 McFarland standard (equivalent to 1.5 × 10^8^ CFU/mL). Subsequently, 100 µL of the inoculum was added to the deer thigh flesh samples. To ensure uniform distribution of the pathogen, the deer flesh samples were thoroughly mixed for three minutes at room temperature after inoculation with *L. monocytogenes*.

### 4.3. Essential Oil Characteristic

The black pepper (*Piper nigrum* L.) berry powder was purchased from the market; the plant material was imported from India. This plant material was hydrodistilled for four hours to extract the volatile fraction. The main component was found to be sesquiterpene (E)-caryophyllene (25.9%), which was followed by large amounts of the monoterpene’s limonene (12.1%) and sabinene (14.5%). In addition, significant concentrations of β-pinene (7.6%), δ-3-carene (6.1%), α-pinene (5.9%), α-thujene (3.0%), and β-phellandrene (2.8%) monoterpene hydrocarbons were found. Furthermore, α-humulene (2.0%), a sesquiterpene hydrocarbon, and 4-terpineol (2.1%), a member of the monoterpene alcohol class, were discovered. The chemical composition of PNEO has been previously published by Vuković et al. [65].

### 4.4. Microbial Analyses

Microbiological evaluations were conducted on specific days (1, 7, 14) throughout the experiment. Before subjecting the samples to heating, they were stored at 4 °C for 24 h. Following heating, portions of the samples were assessed at designated intervals. Initially, 5 g of red deer meat samples were placed into sterile stomacher bags and diluted to 10^−1^, with the addition of 45 mL of peptone water. These samples were then homogenized for 20 min using a stomacher apparatus. Subsequently, 0.1 mL aliquots from appropriate dilutions were spread onto a standard pre-dried plate count agar medium. For the cultivation of coliform bacteria, Violet Red Bile Lactose Agar (VRBL, Oxoid, Basingstoke, UK) was utilized and incubated at 37 °C for 24 to 48 h. Plate Count Agar (PCA, Oxoid, Basingstoke, UK) was employed for the Total Viable Count (TVC, Oxoid, Basingstoke, UK) and incubated at 30 °C for 48 to 72 h. Viable counts were calculated based on growth on this medium. Oxford Agar supplemented with Oxford supplement was used for the enumeration of L. monocytogenes, with plates incubated at 37 °C for 24 h to facilitate bacterial growth.

### 4.5. Identification of Microorganisms Using Mass Spectrometry

Microorganisms obtained from deer thigh flesh samples underwent identification using the MALDI-TOF (Matrix-Assisted Laser Desorption/Ionization Time of Flight) MS Biotyper system from Bruker Daltonics in Bremen, Germany, alongside reference libraries. To create the matrix solution, a stock solution was initially prepared, transitioning into an organic substance. The standard solution consisted of 50% acetonitrile, 47.5% water, and 2.5% trifluoroacetic acid. The stock solution was formulated by combining 500 µL of pure 100% acetonitrile with 475 µL of filtered water and 25 µL of pure 10% trifluoroacetic acid. The “HCCA matrix portioned” was then prepared in a 250 µL Eppendorf flask and mixed with the organic solvent. Matrix materials were procured from Aloqence Science in Vrable, Slovakia, following prior recommendations [66]. Eight different colonies selected from the Petri plate were processed, with biological material from these colonies being transferred to an Eppendorf flask along with 300 µL of distilled water, mixed, and centrifuged for two minutes at 10,000× *g* using a ROTOFIX 32A centrifuge from Ites in Vranov, Slovakia. Subsequently, 900 µL of ethanol was added. Following the removal of the supernatant, the precipitate was dried at an ambient temperature (20 °C). Next, 30 µL of 70% formic acid and 30 µL of acetonitrile were added to the particle. For identification, scores were interpreted as follows: a score below 1.700 was considered unreliable, a score between 2.300 and 3.000 indicated extremely probable species identification, a score between 2.000 and 2.299 suggested genus identification with potential species identification, and a score between 1.700 and 1.999 indicated a likely genus identification.

### 4.6. Statistic Analysis

All assessments were conducted in triplicate, and the results are presented as mean values ± standard deviation (SD). The significance differences among the means were determined by one-way ANOVA (CoStat version 6.451, CoHort Software, Pacific Grove, CA, USA) and Duncan’s MRT; *p* ≤ 0.05 significance was used for the separation of the samples.

Graphic elaboration was performed using a JMP Pro 17.0 software package (SAS Institute, Cary, NC, USA).

## 5. Conclusions

Our study highlighted how temperature, cooking duration, and the addition of black pepper essential oil (PNEO) can significantly reduce the presence of *L. monocytogenes* in sous-vide-cooked red deer meat. This underscores the importance of precise sous-vide cooking techniques for ensuring food safety and minimizing the risk of pathogen contamination. Following recommended time and temperature guidelines during sous-vide cooking is crucial to prevent pathogen survival and growth. Additionally, our investigation revealed a diverse microbiome in sous-vide-cooked red deer meat through mass spectrometry analysis. Alongside *L. monocytogenes*, which was intentionally introduced, we identified several other common species like *Kocuria salsicia*, *Pseudomonas taetrolens*, and *P. fragi*. Storing game meat that has been cooked sous-vide and refrigerated with PNEO can enhance safety. We found that using PNEO at a concentration of 1% had a positive impact on reducing *L. monocytogenes*. Moreover, the antimicrobial effectiveness of PNEO sous-vide red deer meat increased over 14 days of refrigerated storage at 4 °C. The amount of used PNEO can also influence its antimicrobial properties. These findings offer valuable insights for food producers, suggesting that sous-vide cooking with black pepper EO can naturally inhibit the growth of *L. monocytogenes*, making game meat safer for storage at appropriate temperatures.

## Figures and Tables

**Figure 1 molecules-29-04179-f001:**
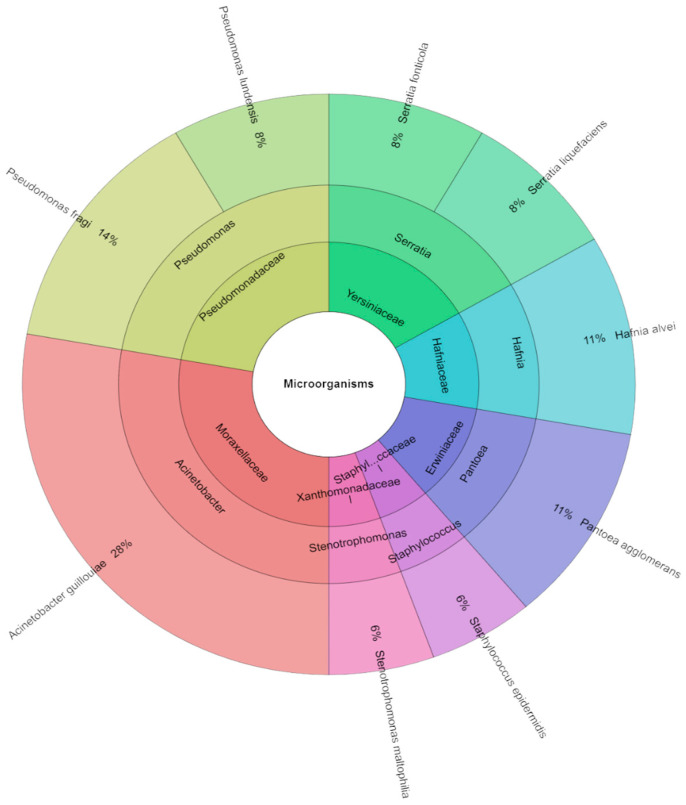
The Krona chart of microorganism species, genera, and families isolated from deer flesh meat on day 0.

**Figure 2 molecules-29-04179-f002:**
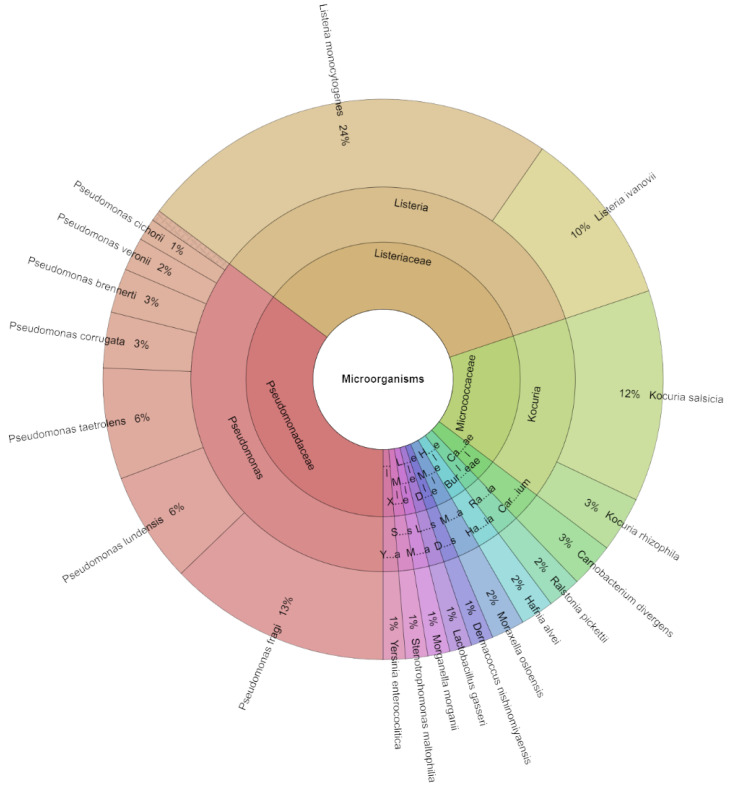
The Krona chart of microorganism species, genera and family isolated from deer flesh meat on day 1.

**Figure 3 molecules-29-04179-f003:**
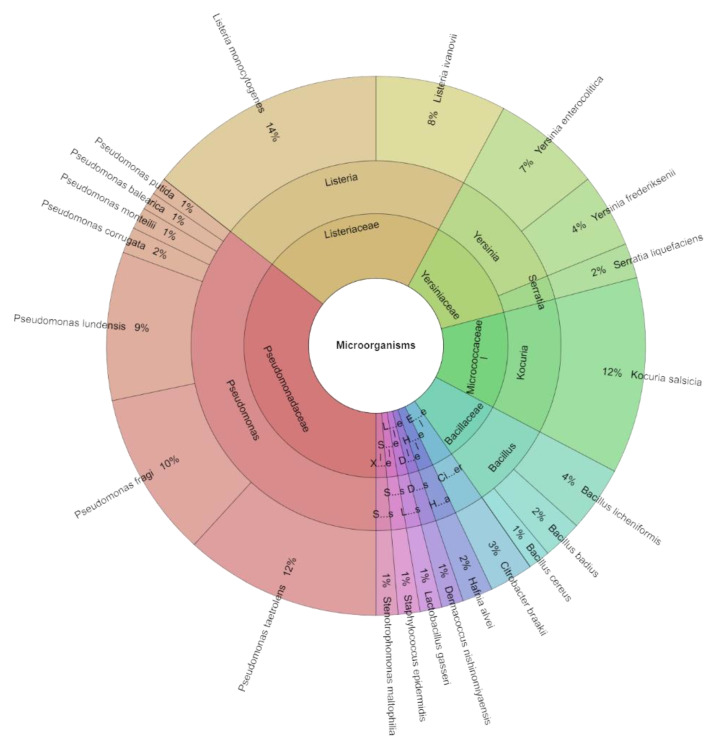
The Krona chart of microorganism species, genera and family isolated from deer flesh meat on day 7.

**Figure 4 molecules-29-04179-f004:**
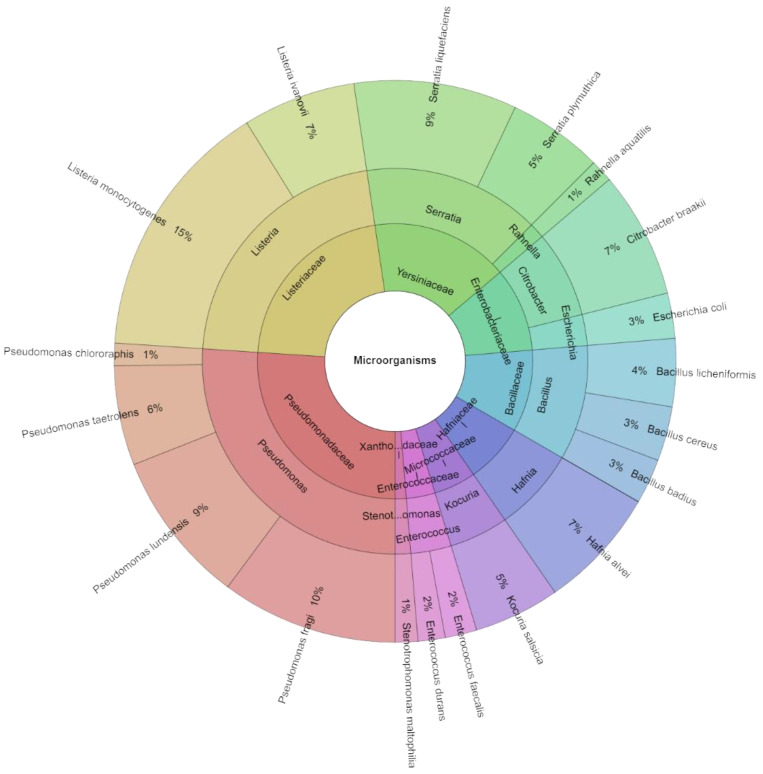
The Krona chart of microorganism species, genera, and families isolated from deer flesh meat on day 14.

**Table 1 molecules-29-04179-t001:** Total viable count (log CFU/g) of sous-vide red deer meat samples after storage 1, 7, and 14 days treated in a water bath at temperatures between 50 and 65 °C for 5 to 20 min. Data are the mean (±SD) of three red deer meat samples. Control: red deer meat samples placed in polyethylene bags without vacuum. Control vacuum: red deer meat samples vacuum-packed in polyethylene bags. Essential oil: deer meat samples treated with 1% PNEO and vacuum-packed. *Listeria monocytogenes*: red deer meat samples inoculated with *L. monocytogenes* and vacuum-packed. Essential oil + *Listeria monocytogenes:* red deer meat samples treated with 1% PNEO and inoculated with *L. monocytogenes* and vacuum-packed.

Treatment	Temperature (°C)	Time (min)	Total Count of Bacteria (log CFU/g)		
			Day of Storage		
			1	7	14
Control	50	5	3.52 ± 0.03 ^b,A^	3.79 ± 0.09 ^a,A^	3.88 ± 0.11 ^a,A^
		10	3.40 ± 0.07 ^b,B^	3.69 ± 0.07 ^a,A^	3.78 ± 0.07 ^a,A^
		15	3.29 ± 0.03 ^c,C^	3.48 ± 0.04 ^b,B^	3.62 ± 0.06 ^a,B^
		20	3.24 ± 0.04 ^b,C^	3.35 ± 0.09 ^ab,BC^	3.52 ± 0.06 ^a,BC^
	55	5	3.17 ± 0.04 ^b,D^	3.33 ± 0.08 ^ab,C^	3.44 ± 0.11 ^a,C^
		10	3.06 ± 0.04 ^b,E^	3.23 ± 0.12 ^a,C^	3.40 ± 0.05 ^a,C^
		15	2.62 ± 0.06 ^c,F^	2.87 ± 0.12 ^b,D^	3.09 ± 0.04 ^a,D^
		20	2.39 ± 0.06 ^c,G^	2.62 ± 0.06 ^b,E^	2.85 ± 0.06 ^a,E^
	60	5	2.18 ± 0.07 ^b,H^	2.45 ± 0.09 ^a,E^	2.65 ± 0.12 ^a,EF^
		10	n.d. ^c,I^	2.25 ± 0.08 ^b,F^	2.45 ± 0.09 ^a,F^
		15	n.d. ^c,I^	1.87 ± 0.09 ^b,G^	2.24 ± 0.11 ^a,G^
		20	n.d. ^c,I^	1.56 ± 0.12 ^b,H^	1.83 ± 0.15 ^a,H^
	65	5	n.d. ^c,I^	1.22 ± 0.10 ^b,I^	1.66 ± 0.10 ^a,HI^
		10	n.d. ^b,I^	n.d. ^b,J^	1.55 ± 0.12 ^a,I^
		15	n.d. ^b,I^	n.d. ^b,J^	1.38 ± 0.04 ^a,J^
		20	n.d. ^b,I^	n.d. ^b,J^	1.20 ± 0.07 ^a,K^
Control vacuum	50	5	3.43 ± 0.02 ^c,A^	3.54 ± 0.08 ^b,A^	3.71 ± 0.06 ^a,A^
		10	3.18 ± 0.06 ^b,B^	3.35 ± 0.11 ^b,A^	3.58 ± 0.06 ^a,B^
		15	2.69 ± 0.03 ^c,C^	2.87 ± 0.10 ^b,B^	3.20 ± 0.03 ^a,C^
		20	2.41 ± 0.06 ^b,D^	2.63 ± 0.16 ^ab,B^	2.77 ± 0.11 ^a,D^
	55	5	2.34 ± 0.12 ^b,D^	2.41 ± 0.03 ^b,C^	2.57 ± 0.03 ^a,E^
		10	2.17 ± 0.05 ^b,E^	2.39 ± 0.05 ^a,C^	2.47 ± 0.06 ^a,F^
		15	2.07 ± 0.02 ^c,F^	2.23 ± 0.04 ^b,D^	2.33 ± 0.05 ^a,G^
		20	1.86 ± 0.09 ^b,G^	2.14 ± 0.06 ^a,E^	2.27 ± 0.16 ^a,GH^
	60	5	n.d. ^b,H^	1.87 ± 0.10 ^a,F^	2.05 ± 0.07 ^a,H^
		10	n.d. ^b,H^	n.d. ^b,G^	1.87 ± 0.11 ^a,H^
		15	n.d. ^b,H^	n.d. ^b,G^	1.55 ± 0.12 ^a,I^
		20	n.d. ^b,H^	n.d. ^b^	1.33 ± 0.10 ^a,IJ^
	65	5	n.d. ^b,H^	n.d. ^b,G^	1.16 ± 0.07 ^a,J^
		10	n.d. ^a,H^	n.d. ^a,G^	n.d. ^a,K^
		15	n.d. ^a,H^	n.d. ^a,G^	n.d. ^a,K^
		20	n.d. ^a,H^	n.d. ^a,G^	n.d. ^a,K^
Essential oil	50	5	3.20 ± 0.06 ^c,A^	3.40 ± 0.06 ^b,A^	3.60 ± 0.14 ^a,A^
		10	3.14 ± 0.06 ^b,A^	3.31 ± 0.04 ^a,A^	3.39 ± 0.06 ^a,B^
		15	2.93 ± 0.06 ^b,B^	3.04 ± 0.08 ^b,B^	3.33 ± 0.12 ^a,BC^
		20	2.71 ± 0.04 ^c,C^	2.87 ± 0.09 ^b,BC^	3.18 ± 0.06 ^a,C^
	55	5	2.49 ± 0.06 ^b,D^	2.78 ± 0.06 ^a,CD^	2.91 ± 0.07 ^a,D^
		10	2.63 ± 0.12 ^a,CD^	2.66 ± 0.10 ^a,D^	2.76 ± 0.09 ^a,D^
		15	2.20 ± 0.03 ^c,E^	2.35 ± 0.09 ^b,E^	2.53 ± 0.04 ^a,E^
		20	1.84 ± 0.07 ^c,F^	2.14 ± 0.07 ^b,F^	2.37 ± 0.04 ^a,F^
	60	5	n.d. ^c,G^	1.77 ± 0.12 ^b,G^	2.23 ± 0.08 ^a,G^
		10	n.d. ^b,G^	n.d. ^b,H^	2.06 ± 0.07 ^a,H^
		15	n.d. ^b,G^	n.d. ^b,H^	1.52 ± 0.12 ^a,I^
		20	n.d. ^b,G^	n.d. ^b,H^	1.38 ± 0.14 ^a,I^
	65	5	n.d. ^a,G^	n.d. ^a,H^	n.d. ^a,J^
		10	n.d. ^a,G^	n.d. ^a,H^	n.d. ^a,J^
		15	n.d. ^a,G^	n.d. ^a,H^	n.d. ^a,J^
		20	n.d. ^a,G^	n.d. ^a,H^	n.d. ^a,J^
*Listeria monocytogenes*	50	5	3.45 ± 0.10 ^b,A^	3.77 ± 0.10 ^a,A^	3.88 ± 0.10 ^a,A^
		10	3.41 ± 0.04 ^b,A^	3.65 ± 0.11 ^ab,AB^	3.74 ± 0.08 ^a,AB^
		15	3.16 ± 0.05 ^b,B^	3.54 ± 0.08 ^a,B^	3.66 ± 0.11 ^a,B^
		20	2.77 ± 0.13 ^c,C^	3.24 ± 0.09 ^b,C^	3.41 ± 0.06 ^a,C^
	55	5	2.38 ± 0.04 ^c,D^	2.87 ± 0.11 ^b,D^	3.17 ± 0.06 ^a,D^
		10	2.18 ± 0.07 ^c,E^	2.45 ± 0.11 ^b,E^	2.80 ± 0.06 ^a,E^
		15	2.00 ± 0.06 ^c,F^	2.34 ± 0.10 ^b,E^	2.52 ± 0.07 ^a,F^
		20	1.87 ± 0.10 ^b,FG^	2.07 ± 0.11 ^b,F^	2.41 ± 0.12 ^a,F^
	60	5	1.73 ± 0.05 ^c,G^	1.96 ± 0.08 ^b,F^	2.12 ± 0.04 ^a,G^
		10	n.d. ^b,H^	n.d. ^b,G^	2.06 ± 0.07 ^a,G^
		15	n.d. ^b,H^	n.d. ^b,G^	1.82 ± 0.06 ^a,H^
		20	n.d. ^b,H^	n.d. ^b,G^	1.72 ± 0.04 ^a,I^
	65	5	n.d. ^b,H^	n.d. ^b,G^	1.34 ± 0.12 ^a,JK^
		10	n.d. ^b,H^	n.d. ^b,G^	1.39 ± 0.14 ^a,J^
		15	n.d. ^b,H^	n.d. ^b,G^	1.17 ± 0.05 ^a,K^
		20	n.d. ^b,H^	n.d. ^b,G^	1.09 ± 0.02 ^a,L^
Essential oil + *Listeria monocytogenes*	50	5	3.44 ± 0.02 ^b,A^	3.66 ± 0.11 ^a,A^	3.80 ± 0.12 ^a,A^
		10	3.23 ± 0.03 ^b,B^	3.56 ± 0.11 ^a,A^	3.70 ± 0.05 ^a,A^
		15	2.78 ± 0.07 ^b,C^	3.22 ± 0.10 ^a,B^	3.43 ± 0.11 ^a,B^
		20	2.57 ± 0.04 ^c,D^	2.88 ± 0.10 ^b,C^	3.28 ± 0.13 ^a,B^
	55	5	2.43 ± 0.04 ^b,E^	2.76 ± 0.13 ^a,C^	2.86 ± 0.10 ^a,C^
		10	2.07 ± 0.11 ^c,F^	2.45 ± 0.11 ^b,D^	2.71 ± 0.05 ^a,C^
		15	1.92 ± 0.07 ^c,FG^	2.25 ± 0.07 ^b,E^	2.49 ± 0.14 ^a,D^
		20	1.86 ± 0.09 ^b,G^	1.97 ± 0.03 ^b,F^	2.32 ± 0.03 ^a,D^
	60	5	n.d. ^b,H^	n.d. ^b,G^	2.13 ± 0.07 ^a,E^
		10	n.d. ^b,H^	n.d. ^b,G^	1.87 ± 0.11 ^a,F^
		15	n.d. ^b,H^	n.d. ^b,G^	1.78 ± 0.07 ^a,F^
		20	n.d. ^b,H^	n.d. ^b,G^	1.64 ± 0.04 ^a,G^
	65	5	n.d. ^b,H^	n.d. ^b,G^	1.22 ± 0.02 ^a,H^
		10	n.d. ^b,H^	n.d. ^b,G^	1.19 ± 0.06 ^a,H^
		15	n.d. ^b,H^	n.d. ^b,G^	1.12 ± 0.06 ^a,I^
		20	n.d. ^b,H^	n.d. ^b,G^	1.03 ± 0.03 ^a,I^

^a–c^ Different superscript lowercase letters indicate statistically different values within row (Duncan’s MRT, *p* ≤ 0.05). ^A–L^ Different superscript uppercase letters indicate statistically different values within the column for each treatment (Duncan’s MRT, *p* ≤ 0.05). n.d. = not detected (value = 0.00).

**Table 2 molecules-29-04179-t002:** Total coliforms bacteria (log CFU/g) of sous-vide red deer meat samples after storage 1, 7, and 14 days treated in a water bath at temperatures between 50 and 65 °C for 5 to 20 min. Data are the mean (±SD) of three red deer meat samples. Control: red deer meat samples placed in polyethylene bags without vacuum. Control vacuum: red deer meat samples vacuum-packed in polyethylene bags. Essential oil: red deer meat samples treated with 1% PNEO and vacuum-packed. *Listeria monocytogenes*: red deer meat samples inoculated with *L. monocytogenes* and vacuum-packed. Essential oil + *Listeria monocytogenes:* red deer meat samples treated with 1% PNEO and inoculated with *L. monocytogenes* and vacuum-packed.

Treatment	Temperature (°C)	Time (min)	Coliforms Bacteria (log CFU/g)		
			Day		
			1	7	14
Control	50	5	n.d. ^c,A^	3.61 ± 0.04 ^b,A^	3.77 ± 0.10 ^a,A^
		10	n.d. ^c,A^	3.46 ± 0.06 ^b,B^	3.67 ± 0.11 ^a,A^
		15	n.d. ^b,A^	3.25 ± 0.08 ^a,C^	3.37 ± 0.12 ^a,B^
		20	n.d. ^c,A^	2.87 ± 0.10 ^b,D^	3.26 ± 0.17 ^a,B^
	55	5	n.d. ^c,A^	2.52 ± 0.06 ^b,E^	2.81 ± 0.16 ^a,C^
		10	n.d. ^b,A^	2.25 ± 0.07 ^a,F^	2.60 ± 0.23 ^a,C^
		15	n.d. ^a,A^	n.d. ^a,G^	n.d. ^a,D^
		20	n.d. ^a,A^	n.d. ^a,G^	n.d. ^a,D^
	60	5	n.d. ^a,A^	n.d. ^a,G^	n.d. ^a,D^
		10	n.d. ^a,A^	n.d. ^a,G^	n.d. ^a,D^
		15	n.d. ^a,A^	n.d. ^a,G^	n.d. ^a,D^
		20	n.d. ^a,A^	n.d. ^a,G^	n.d. ^a,D^
	65	5	n.d. ^a,A^	n.d. ^a,G^	n.d. ^a,D^
		10	n.d. ^a,A^	n.d. ^a,G^	n.d. ^a,D^
		15	n.d. ^a,A^	n.d. ^a,G^	n.d. ^a,D^
		20	n.d. ^a,A^	n.d. ^a,G^	n.d. ^a,D^
Control vacuum	50	5	n.d. ^b,A^	2.87 ± 0.11 ^a,A^	3.08 ± 0.17 ^a,A^
		10	n.d. ^c,A^	2.45 ± 0.10 ^b,B^	2.81 ± 0.16 ^a,A^
		15	n.d. ^b,A^	1.86 ± 0.12 ^a,C^	2.11 ± 0.13 ^a,B^
		20	n.d. ^c,A^	1.54 ± 0.10 ^b,D^	1.77 ± 0.12 ^a,C^
	55	5	n.d. ^c,A^	1.33 ± 0.09 ^b,E^	1.58 ± 0.06 ^a,D^
		10	n.d. ^a,A^	n.d. ^a,F^	n.d. ^a,E^
		15	n.d. ^a,A^	n.d. ^a,F^	n.d. ^a,E^
		20	n.d. ^a,A^	n.d. ^a,F^	n.d. ^a,E^
	60	5	n.d. ^a,A^	n.d. ^a,F^	n.d. ^a,E^
		10	n.d. ^a,A^	n.d. ^a,F^	n.d. ^a,E^
		15	n.d. ^a,A^	n.d. ^a,F^	n.d. ^a,E^
		20	n.d. ^a,A^	n.d. ^a,F^	n.d. ^a,E^
	65	5	n.d. ^a,A^	n.d. ^a,F^	n.d. ^a,E^
		10	n.d. ^a,A^	n.d. ^a,F^	n.d. ^a,E^
		15	n.d. ^a,A^	n.d. ^a,F^	n.d. ^a,E^
		20	n.d. ^a,A^	n.d. ^a,F^	n.d. ^a,E^
Essential oil	50	5	n.d. ^b,A^	2.34 ± 0.11 ^a,A^	2.55 ± 0.22 ^a,A^
		10	n.d. ^c,A^	1.14 ± 0.08 ^b,B^	1.80 ± 0.17 ^a,B^
		15	n.d. ^b,A^	n.d. ^b,C^	1.59 ± 0.16 ^a,B^
		20	n.d. ^a,A^	n.d. ^a,C^	n.d. ^a,C^
	55	5	n.d. ^a,A^	n.d. ^a,C^	n.d. ^a,C^
		10	n.d. ^a,A^	n.d. ^a,C^	n.d. ^a,C^
		15	n.d. ^a,A^	n.d. ^a,C^	n.d. ^a,C^
		20	n.d. ^a,A^	n.d. ^a,C^	n.d. ^a,C^
	60	5	n.d. ^a,A^	n.d. ^a,C^	n.d. ^a,C^
		10	n.d. ^a,A^	n.d. ^a,C^	n.d. ^a,C^
		15	n.d. ^a,A^	n.d. ^a,C^	n.d. ^a,C^
		20	n.d. ^a,A^	n.d. ^a,C^	n.d. ^a,C^
	65	5	n.d. ^a,A^	n.d. ^a,C^	n.d. ^a,C^
		10	n.d. ^a,A^	n.d. ^a,C^	n.d. ^a,C^
		15	n.d. ^a,A^	n.d. ^a,C^	n.d. ^a,C^
		20	n.d. ^a,A^	n.d. ^a,C^	n.d. ^a,C^
*Listeria monocytogenes*	50	5	n.d. ^c,A^	2.21 ± 0.08 ^b,A^	3.12 ± 0.13 ^a,A^
		10	n.d. ^c,A^	1.87 ± 0.11 ^b,B^	2.78 ± 0.11 ^a,B^
		15	n.d. ^b,A^	n.d. ^b,C^	2.24 ± 0.09 ^a,C^
		20	n.d. ^b,A^	n.d. ^b,C^	1.87 ± 0.11 ^a,D^
	55	5	n.d. ^a,A^	n.d. ^a,C^	n.d. ^a,E^
		10	n.d. ^a,A^	n.d. ^a,C^	n.d. ^a,E^
		15	n.d. ^a,A^	n.d. ^a,C^	n.d. ^a,E^
		20	n.d. ^a,A^	n.d. ^a,C^	n.d. ^a,E^
	60	5	n.d. ^a,A^	n.d. ^a,C^	n.d. ^a,E^
		10	n.d. ^a,A^	n.d. ^a,C^	n.d. ^a,E^
		15	n.d. ^a,A^	n.d. ^a,C^	n.d. ^a,E^
		20	n.d. ^a,A^	n.d. ^a,C^	n.d. ^a,E^
	65	5	n.d. ^a,A^	n.d. ^a,C^	n.d. ^a,E^
		10	n.d. ^a,A^	n.d. ^a,C^	n.d. ^a,E^
		15	n.d. ^a,A^	n.d. ^a,C^	n.d. ^a,E^
		20	n.d. ^a,A^	n.d. ^a,C^	n.d. ^a,E^
Essential oil + *Listeria monocytogenes*	50	5	n.d. ^c,A^	1.86 ± 0.12 ^b,A^	2.31 ± 0.09 ^a,A^
		10	n.d. ^b,A^	n.d. ^b,B^	2.20 ± 0.23 ^a,A^
		15	n.d. ^b,A^	n.d. ^b,B^	1.77 ± 0.11 ^a,B^
		20	n.d. ^a,A^	n.d. ^a,B^	n.d. ^a,C^
	55	5	n.d. ^a,A^	n.d. ^a,B^	n.d. ^a,C^
		10	n.d. ^a,A^	n.d. ^a,B^	n.d. ^a,C^
		15	n.d. ^a,A^	n.d. ^a,B^	n.d. ^a,C^
		20	n.d. ^a,A^	n.d. ^a,B^	n.d. ^a,C^
	60	5	n.d. ^a,A^	n.d. ^a,B^	n.d. ^a,C^
		10	n.d. ^a,A^	n.d. ^a,B^	n.d. ^a,C^
		15	n.d. ^a,A^	n.d. ^a,B^	n.d. ^a,C^
		20	n.d. ^a,A^	n.d. ^a,B^	n.d. ^a,C^
	65	5	n.d. ^a,A^	n.d. ^a,B^	n.d. ^a,C^
		10	n.d. ^a,A^	n.d. ^a,B^	n.d. ^a,C^
		15	n.d. ^a,A^	n.d. ^a,B^	n.d. ^a,C^
		20	n.d. ^a,A^	n.d. ^a,B^	n.d. ^a,C^

^a–c^ Different superscript lowercase letters indicate statistically different values within row (Duncan’s MRT, *p* ≤ 0.05). ^A–G^ Different superscript uppercase letters indicate statistically different values within columns for each treatment (Duncan’s MRT, *p* ≤ 0.05). n.d. = not detected (value = 0.00).

**Table 3 molecules-29-04179-t003:** *L. monocytogenes* count (log CFU/g) of sous-vide red deer meat samples after storage 1, 7, and 14 days treated in a water bath at temperatures between 50 and 65 °C for 5 to 20 min. Data are the mean (±SD) of three red deer meat samples. Control: red deer meat samples placed in polyethylene bags without vacuum. Control vacuum: red deer meat samples vacuum-packed in polyethylene bags. Essential oil: red deer meat samples treated with 1% PNEO and vacuum-packed. *Listeria monocytogenes*: red deer meat samples inoculated with *L. monocytogenes* and vacuum-packed. Essential oil + *Listeria monocytogenes:* red deer meat samples treated with 1% PNEO and inoculated with *L. monocytogenes* and vacuum-packed.

	Temperature (°C)	Time (min)	*Listeria monocytogenes* (log CFU/g)		
			Day		
			1	7	14
Control	50	5	n.d. ^a,A^	n.d. ^a,A^	n.d. ^a,A^
		10	n.d. ^a,A^	n.d. ^a,A^	n.d. ^a,A^
		15	n.d. ^a,A^	n.d. ^a,A^	n.d. ^a,A^
		20	n.d. ^a,A^	n.d. ^a,A^	n.d. ^a,A^
	55	5	n.d. ^a,A^	n.d. ^a,A^	n.d. ^a,A^
		10	n.d. ^a,A^	n.d. ^a,A^	n.d. ^a,A^
		15	n.d. ^a,A^	n.d. ^a,A^	n.d. ^a,A^
		20	n.d. ^a,A^	n.d. ^a,A^	n.d. ^a,A^
	60	5	n.d. ^a,A^	n.d. ^a,A^	n.d. ^a,A^
		10	n.d. ^a,A^	n.d. ^a,A^	n.d. ^a,A^
		15	n.d. ^a,A^	n.d. ^a,A^	n.d. ^a,A^
		20	n.d. ^a,A^	n.d. ^a,A^	n.d. ^a,A^
	65	5	n.d. ^a,A^	n.d. ^a,A^	n.d. ^a,A^
		10	n.d. ^a,A^	n.d. ^a,A^	n.d. ^a,A^
		15	n.d. ^a,A^	n.d. ^a,A^	n.d. ^a,A^
		20	n.d. ^a,A^	n.d. ^a,A^	n.d. ^a,A^
Control vacuum	50	5	n.d. ^a,A^	n.d. ^a,A^	n.d. ^a,A^
		10	n.d. ^a,A^	n.d. ^a,A^	n.d. ^a,A^
		15	n.d. ^a,A^	n.d. ^a,A^	n.d. ^a,A^
		20	n.d. ^a,A^	n.d. ^a,A^	n.d. ^a,A^
	55	5	n.d. ^a,A^	n.d. ^a,A^	n.d. ^a,A^
		10	n.d. ^a,A^	n.d. ^a,A^	n.d. ^a,A^
		15	n.d. ^a,A^	n.d. ^a,A^	n.d. ^a,A^
		20	n.d. ^a,A^	n.d. ^a,A^	n.d. ^a,A^
	60	5	n.d. ^a,A^	n.d. ^a,A^	n.d. ^a,A^
		10	n.d. ^a,A^	n.d. ^a,A^	n.d. ^a,A^
		15	n.d. ^a,A^	n.d. ^a,A^	n.d. ^a,A^
		20	n.d. ^a,A^	n.d. ^a,A^	n.d. ^a,A^
	65	5	n.d. ^a,A^	n.d. ^a,A^	n.d. ^a,A^
		10	n.d. ^a,A^	n.d. ^a,A^	n.d. ^a,A^
		15	n.d. ^a,A^	n.d. ^a,A^	n.d. ^a,A^
		20	n.d. ^a,A^	n.d. ^a,A^	n.d. ^a,A^
Essential oil	50	5	n.d. ^a,A^	n.d. ^a,A^	n.d. ^a,A^
		10	n.d. ^a,A^	n.d. ^a,A^	n.d. ^a,A^
		15	n.d. ^a,A^	n.d. ^a,A^	n.d. ^a,A^
		20	n.d. ^a,A^	n.d. ^a,A^	n.d. ^a,A^
	55	5	n.d. ^a,A^	n.d. ^a,A^	n.d. ^a,A^
		10	n.d. ^a,A^	n.d. ^a,A^	n.d. ^a,A^
		15	n.d. ^a,A^	n.d. ^a,A^	n.d. ^a,A^
		20	n.d. ^a,A^	n.d. ^a,A^	n.d. ^a,A^
	60	5	n.d. ^a,A^	n.d. ^a,A^	n.d. ^a,A^
		10	n.d. ^a,A^	n.d. ^a,A^	n.d. ^a,A^
		15	n.d. ^a,A^	n.d. ^a,A^	n.d. ^a,A^
		20	n.d. ^a,A^	n.d. ^a,A^	n.d. ^a,A^
	65	5	n.d. ^a,A^	n.d. ^a,A^	n.d. ^a,A^
		10	n.d. ^a,A^	n.d. ^a,A^	n.d. ^a,A^
		15	n.d. ^a,A^	n.d. ^a,A^	n.d. ^a,A^
		20	n.d. ^a,A^	n.d. ^a,A^	n.d. ^a,A^
*Listeria monocytogenes*	50	5	3.17 ± 0.05 ^a,A^	2.83 ± 0.05 ^b,A^	3.10 ± 0.12 ^a,A^
		10	2.45 ± 0.11 ^a,B^	2.44 ± 0.11 ^a,B^	2.55 ± 0.10 ^a,B^
		15	2.20 ± 0.08 ^ab,C^	2.06 ± 0.07 ^b,C^	2.24 ± 0.09 ^a,C^
		20	n.d. ^b,D^	1.74 ± 0.06 ^a,D^	1.91 ± 0.12 ^a,D^
	55	5	n.d. ^b,D^	1.28 ± 0.06 ^a,E^	1.37 ± 0.07 ^a,E^
		10	n.d. ^a,D^	n.d. ^a,F^	n.d. ^a,F^
		15	n.d. ^a,D^	n.d. ^a,F^	n.d. ^a,F^
		20	n.d. ^a,D^	n.d. ^a,F^	n.d. ^a,F^
	60	5	n.d. ^a,D^	n.d. ^a,F^	n.d. ^a,F^
		10	n.d. ^a,D^	n.d. ^a,F^	n.d. ^a,F^
		15	n.d. ^a,D^	n.d. ^a,F^	n.d. ^a,F^
		20	n.d. ^a,D^	n.d. ^a,F^	n.d. ^a,F^
	65	5	n.d. ^a,D^	n.d. ^a,F^	n.d. ^a,F^
		10	n.d. ^a,D^	n.d. ^a,F^	n.d. ^a,F^
		15	n.d. ^a,D^	n.d. ^a,F^	n.d. ^a,F^
		20	n.d. ^a,D^	n.d. ^a,F^	n.d. ^a,F^
Essential oil + *Listeria monocytogenes*	50	5	2.89 ± 0.08 ^a,A^	2.56 ± 0.11 ^b,a^	2.69 ± 0.16 ^ab,A^
		10	2.32 ± 0.14 ^ab,B^	2.20 ± 0.06 ^b,B^	2.37 ± 0.07 ^a,B^
		15	n.d. ^b,C^	1.97 ± 0.19 ^a,B^	2.24 ± 0.17 ^a,B^
		20	n.d. ^a,C^	n.d. ^a,C^	n.d. ^a,C^
	55	5	n.d. ^a,C^	n.d. ^a,C^	n.d. ^a,C^
		10	n.d. ^a,C^	n.d. ^a,C^	n.d. ^a,C^
		15	n.d. ^a,C^	n.d. ^a,C^	n.d. ^a,C^
		20	n.d. ^a,C^	n.d. ^a,C^	n.d. ^a,C^
	60	5	n.d. ^a,C^	n.d. ^a,C^	n.d. ^a,C^
		10	n.d. ^a,C^	n.d. ^a,C^	n.d. ^a,C^
		15	n.d. ^a,C^	n.d. ^a,C^	n.d. ^a,C^
		20	n.d. ^a,C^	n.d. ^a,C^	n.d. ^a,C^
	65	5	n.d. ^a,C^	n.d. ^a,C^	n.d. ^a,C^
		10	n.d. ^a,C^	n.d. ^a,C^	n.d. ^a,C^
		15	n.d. ^a,C^	n.d. ^a,C^	n.d. ^a,C^
		20	n.d. ^a,C^	n.d. ^a,C^	n.d. ^a,C^

^a–b^ Different superscript lowercase letters indicate statistically different values within row (Duncan’s MRT, *p* ≤ 0.05). ^A–F^ Different superscript uppercase letters indicate statistically different values within the column for each treatment (Duncan’s MRT, *p* ≤ 0.05). n.d. = not detected (value = 0.00).

## Data Availability

Data will be made available on request.

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
