# Peer review of "Enhancing the Shelf Life of Sous-Vide Red Deer Meat with Piper nigrum Essential Oil: A Study on Antimicrobial Efficacy against Listeria monocytogenes"

_molecules, 2024, doi:10.3390/molecules29174179_

Round 1

Reviewer 1 Report (Previous Reviewer 1)

Comments and Suggestions for Authors

The manuscript titled "Characterization and Antimicrobial Power of Piper Nigrum Essential Oil: Application for Extending Shelf Life of Sous-Vide Deer Meat" demonstrates the potential use of sous-vide technology in combination with essential oils to extend food preservation. However, the authors need to clarify the following points:

  1. How does the concentration of black pepper essential oil (PNEO) affect the reduction of L. monocytogenes in sous-vide cooked deer meat?
  2. What are the specific mechanisms by which PNEO inhibits the growth of L. monocytogenes in deer meat?
  3. How does PNEO interact with the structure of the meat or the pathogens at a molecular level to exert its antimicrobial effect?
  4. Does PNEO affect other microorganisms present in deer meat? How does its efficacy against L. monocytogenes compare to its impact on other identified bacteria?
  5. What were the differences in the reduction of L. monocytogenes between the samples treated with PNEO and the control samples without PNEO over the storage period?
  6. What differences in microbial diversity were observed between the samples with and without PNEO during storage?
  7. How did the refrigerated storage time (14 days) influence the efficacy of PNEO in inhibiting L. monocytogenes?
  8. Are there additional factors in the storage process that might influence the efficacy of PNEO, such as the exact storage temperature or interactions with other components of the packaging?
  9. How does the presence of different microbial species in deer meat affect the performance of PNEO as an antimicrobial agent?
  10. What implications do changes in the identified microbiota during storage have for the food safety of the processed product?
  11. How does the efficacy of PNEO in deer meat compare with its efficacy in other types of meat in terms of pathogen control?
  12. What impact does the addition of PNEO have on the organoleptic characteristics of sous-vide cooked deer meat, such as taste, aroma, and texture?
  13. Are there interactions between PNEO and other ingredients or additives used in the processing of deer meat that could affect its antimicrobial efficacy?

Author Response

Reviewer #1

The Authors are very grateful to the Reviewer for their valuable comments. We want to thank the Reviewer for the time devoted to point out constructive and important comments to improve our paper. Based on your questions listed in the review, we present some points in the discussion.

The manuscript titled "Characterization and Antimicrobial Power of Piper Nigrum Essential Oil: Application for Extending Shelf Life of Sous-Vide Deer Meat" demonstrates the potential use of sous-vide technology in combination with essential oils to extend food preservation. However, the authors need to clarify the following points:

Comment 1: How does the concentration of black pepper essential oil (PNEO) affect the reduction of L. monocytogenes in sous-vide cooked deer meat?

Response: In general, the use of EOs as food ingredients has several limitations. Their intense aroma may alter the organoleptic properties of foods, which may be unacceptable for consumers. With this drawback in mind, food processors have to use EOs at low concentrations, which, in turn, due to the possible interaction with food constituents (fats, starch or proteins) and external factors (light, oxidation or heating), may result in poor antimicrobial activity or even inefficiency. Therefore, researchers are looking for alternative ways of use in order to achieve both consumer acceptance and food safety. An effective strategy is to apply EOs in the active packaging of food products. This involves the migration of active compounds from the packaging to the food, providing protection against microorganisms. The appropriate concentration of PNEO has been proposed based on data previously published [1-3] on other types of meat and our own results. The optimal concentration for the application of EOs is between 0.5 and 2%. However, to broaden our research and optimize the experimental part, we are currently working with other EOs and their applications on different types of meat. In our study and compare with other study the EO reduce of L. monocytogenes in sous vide method.

1-Pesavento, G.; Calonico, C.; Bilia, A.R.; Barnabei, M.; Calesini, F.; Addona, R.; Mencarelli, L.; Carmagnini, L.; Di Martino, M.C.; Lo Nostro, A. Antibacterial Activity of Oregano, Rosmarinus and Thymus Essential Oils against Staphylococcus aureus and Listeria monocytogenes in Beef Meatballs. Food Control 201554, 188–199.

2-Gouveia, A.R.; Alves, M.; Silva, J.A.; Saraiva, C. The Antimicrobial Effect of Rosemary and Thyme Essential Oils against Listeria monocytogenes in Sous Vide Cook-Chill Beef During Storage. Procedia Food Sci. 20167, 173–176.

3-Khaleque, M.A.; Keya, C.A.; Hasan, K.N.; Hoque, M.M.; Inatsu, Y.; Bari, M.L. Use of Cloves and Cinnamon Essential Oil to Inactivate Listeria monocytogenes in Ground Beef at Freezing and Refrigeration Temperatures. LWT 201674, 219–223.

Comment 2: What are the specific mechanisms by which PNEO inhibits the growth of L. monocytogenes in deer meat?

Response: The inhibitory mechanisms of PNEO against L. monocytogenes encompassed perturbation of cellular morphology, elevation in reactive oxygen species levels, augmentation of lipid oxidation extent, hyperpolarization of membrane potential, and reduction in intracellular ATP concentration.

Comment 3: How does PNEO interact with the structure of the meat or the pathogens at a molecular level to exert its antimicrobial effect?

Response: Meat is a nutrient dense food item, providing an optimum environment for proliferation of microorganisms and often leads to the spoilage of valuable protein source. Therefore, it is essential that adequate safety measures are applied to maintain its quality. It will definitely ascertain the consumers’ satisfaction from the viewpoint of healthier food. In this context, essential oils provide a new, promising and natural alternative for safety of food against pathogens and micro-organisms. Phenolic compounds in the essential oil are mainly responsible for their anti-microbial activity. However, their use is still limited because they contribute pungent and intense aroma to the food, thus there is need to work out on the novel technologies which can not only increase the food stability butat the same time, are sensorially acceptable to the consumers. High concentrations of fats and/or proteins in foodstuffs may protect bacteria, as they could provide a protective layer and absorb EOs, thus decreasing their concentration and effectiveness in the aqueous phase; on the other hand, high water, and/or salt levels appear to facilitate the action of EOs.

Comment 4: Does PNEO affect other microorganisms present in deer meat? How does its efficacy against L. monocytogenes compare to its impact on other identified bacteria?

Response: PNEO affected other microorganisms in deer meat, so show our results. In the group where PNEO was used the number of microorganisms was lower. The similar results were found in the group where L. monocytogenes inoculated on the meat treated with PNEO.

Comment 5: What were the differences in the reduction of L. monocytogenes between the samples treated with PNEO and the control samples without PNEO over the storage period?

Response: In the experimental group where EO application was not used, the number of L. monocytogenes was higher compared to the groups that were sharpened with EO. The number was lower at the level of approximately 1 log on each monitored day.

Comments 6: What differences in microbial diversity were observed between the samples with and without PNEO during storage?

Response: There was a higher diversity of microorganism species in the control groups without PNEO treatment.

Comments 7:  How did the refrigerated storage time (14 days) influence the efficacy of PNEO in inhibiting L. monocytogenes?

Response: The effectiveness of PNEO during storage was mainly manifested on the 7th day of storage, where the number of L. monocytogenes was also lower compared to the first day. On the 14th day of storage, this number was higher than on the 7th day of storage, but was lower than on the 1st day of storage.

Comments 8:  Are there additional factors in the storage process that might influence the efficacy of PNEO, such as the exact storage temperature or interactions with other components of the packaging?

Response: Packaging fulfils several purposes, including preventing contamination during distribution, preserving product integrity, and maintaining the desired flavor profile of the product.  Unsuitable packaging and storage can induce alterations of EO in the chemical composition of their active substance which may affect the flavor and fragrance properties of the product, with a negative impact on the industrial value and consumer satisfaction. Therefore, determining a suitable packaging method to maintain higher concentrations of active substances during storage is very important.  It has been observed that the composition of essential oils readily changes upon processing and storage of the isolated oil, whereby factors such as temperature, light and oxygen availability make a crucial impact on alteration processes

Comments 9:  How does the presence of different microbial species in deer meat affect the performance of PNEO as an antimicrobial agent?

Response: In our studies, the most frequently isolated species were from the genus Pseudomonas, as well as Serratia, Hafnia and others. From our previous studies as well as other authors, it was found that PNEO has a better antimicrobial effect on gram-positive bacteria. Our results show that PNEO inhibited the growth of microorganisms from the Micrococaccea family, as far as the species representation is concerned.

Comments 10:  What implications do changes in the identified microbiota during storage have for the food safety of the processed product?

Response: Growth of meat microbiota is a dynamic process. Spoilage of meat can be a result of the contiguous growth of different bacteria; thus, quantification of only single bacterial groups may not be sufficient to judge the hygiene status of meat. Different spoilage-related species and strains can colonize the meat surface through different stages involving adsorption to the meat surface and attachment by glycocalix formation. The development of these phases depends on the intrinsic and extrinsic ecological factors of a particular meat ecosystem such as pH, meat surface morphology, O2 availability, temperature, and the presence and development of other bacteria. Many groups of organisms contain members that potentially contribute to meat spoilage under appropriate conditions. This makes the microbial ecology of spoiling raw meat very complex and the spoilage thus very difficult to prevent. Under aerobic conditions, a few species of the genus Pseudomonas are generally recognized to dominate the meat system and to actively contribute to spoilage owing to their capability for glucose and amino acid degradation, even at refrigeration temperatures. Brochothrix thermosphacta is a microorganism for which meat is considered an ecological niche, even though it can also occur in spoiled fish. The capability of B. thermosphacta to grow on meat during both aerobiosis and anaerobiosis makes it a significant meat colonizer and an important member of the spoilage-related flora due to off-odor production. Many members of the Enterobacteriaceae, belonging to the genera Serratia, Enterobacter, Pantoea, Proteus, and Hafnia, often contribute to meat spoilage. Moreover, lactic acid bacteria (LAB) such as Lactobacillus, Carnobacterium, and Leuconostoc can play an important role in the spoilage of refrigerated meat and are also recognized as important competitors of the other spoilage-related microbial groups under appropriate conditions

Comments 11:  How does the efficacy of PNEO in deer meat compare with its efficacy in other types of meat in terms of pathogen control?

Response: A similar antimicrobial effect of PNEO was found e.g. with minced meat, pork. Although the antimicrobial activity in vitro of PNEO has been moderately effective on meat model, the potential use of PNEO in food preservation technologies should be found in optimal concentrations to ensure the safety of the food, appropriated organoleptical characteristics, and accepted by consumers. Studies aiming to elucidate the interaction between essential oils and components of food matrices or additives, stability of oils during food processing, and the standardization of antibacterial methods are still needed.

Comments 12:  What impact does the addition of PNEO have on the organoleptic characteristics of sous-vide cooked deer meat, such as taste, aroma, and texture?

Response: Black pepper (Piper nigrum L.) is one of the main flavoring agents in meat processing. With a relatively high percentage of terpenoids (limonene, α- and β-pinene and caryophyllene), EOs isolated from black pepper (BPEOs), show a strong antioxidant effect, as well as a preservative effect against a broad spectrum of microorganisms. BPEO was added as a natural preservative in fresh pork loin at concentrations of 0 to 0.5%. All batches were stored at 4 °C for 9 days. The study showed that BPEO delayed lipid oxidation and reduced the growth of Enterobacteriaceae and Pseudomonas spp. in fresh pork. In another study, the effect of BPEO coating (0.05 and 0.1%) on lipid oxidation and sensory quality (aroma) of ham was examined. The authors suggested that the use of BPEO has a strong potential to suppress lipid oxidation and improve the sensory acceptability of ham during long-term storage (4 months at room temperature). Overall, the results suggest that BPEO could be used as a natural antioxidant in meat products.

Comments 13:  Are there interactions between PNEO and other ingredients or additives used in the processing of deer meat that could affect its antimicrobial efficacy?

Response: The antimicrobial effect of PNEO is mainly due to their constituent phenolic compounds. The mode of action of PNEO may include damage to cytoplasmic membranes, protein denaturation, cytoplasmic coagulation, and depletion of the proton motive force. The inhibitory effect of PNEO on L. monocytogenes has been documented in numerous studies. The obtained results indicated that LM populations increased during seven and fourteen days of storage at 4 °C in the control groups but decreased when exposed to EOs treatment. At concentrations of 0.5% and 1%, EOs limited the growth of LM in meat at both temperatures, with better effects at the higher dose of 1%. In conclusion, EOs slowed the growth rates of L. monocytogenes populations compared to control during 14 days of storage at 4 °C. Further data show the efficacy of EO (1% v/w) in a meat model against two levels of an L. monocytogenes cocktail (3 and 6 log CFU/g) combined with storage at 4 °C for 14 days.

Reviewer 2 Report (Previous Reviewer 3)

Comments and Suggestions for Authors

Comments to the Authors

The manuscript entitled “Characterization and Antimicrobial Power of Piper Nigrum Essential Oil: Application for Extending Shelf Life of Sous-Vide Deer Meat”. The topic is quite innovative and interesting, the MS is acceptable in the present form.

As a suggestion for subsequent studies, in order to have greater reproducibility in relation to the process, it might be appropriate to use essential oils obtained from the distillation of fresh plant material whose provenance has been certified through botanical identification of the species

Author Response

Reviewer #2

Comments to the Authors

The manuscript entitled “Characterization and Antimicrobial Power of Piper Nigrum Essential Oil: Application for Extending Shelf Life of Sous-Vide Deer Meat”. The topic is quite innovative and interesting, the MS is acceptable in the present form.

As a suggestion for subsequent studies, in order to have greater reproducibility in relation to the process, it might be appropriate to use essential oils obtained from the distillation of fresh plant material whose provenance has been certified through botanical identification of the species

The Authors are very grateful to the Reviewer for their valuable comments. We want to thank the Reviewer for the time devoted to point out constructive and important comments to improve our paper.

Response: We are understanding question, but in this study was conducted commercial essential oil from berry powder. The plants for the study was certified for botanical identification.

Reviewer 3 Report (New Reviewer)

Comments and Suggestions for Authors

The comments in the attachement.

Author Response

Reviewer #3

The Authors are very grateful to the Reviewer for their valuable comments. We want to thank the Reviewer for the time devoted to point out constructive and important comments to improve our paper.

Title: Characterization and antimicrobial Power of Piper Nigrum Essential Oil: application for Extending Shelf Life of Sous-Vide Deer Meat

General review

The study aimed to evaluate the effectiveness of sous-vide and Piper Nigrum essential Oil (PNEO) in preserving red deer meat and reducing microbial contamination. Microbial analysis, including TVC, CB, and L. monocytogenes counts, was conducted on samples subjected to different sous-vide conditions and PNEO treatments. Mass spectrometry was used to identify microbial species. Sous-vide and PNEO reduced microbial counts, with the lowest counts observed in samples treated with 1% PNEO. Kocuria salsicia, Pseudomonas taetrolens, and Pseudomonas fragi were the most frequently isolated microorganisms. The study addresses a relevant topic, employs a controlled experimental design, and demonstrate the antimicrobial effects of the treatments. Meanwhile, the study lacks sufficient sample size and replication, detailed information on sample handling, sensory evaluation, and longer storage period.

Recommendations

To enhance the manuscript, the authors should:

Comments 1: Rewrite the manuscript title to be short and more attractive.

Response: it was done.

Comments 2: Increase sample size and replication for improved statistical power.

Response: In this study were 723 samples used. We thin that number is appreciate for the statistic evaluation. For each temperature and time, and different group was triplicate repetition of samples.

Comments 3: Provide comprehensive details on sample preparation, storage, and inoculation.

Response: It is described in material and method.

Comments 4: Conduct sensory analysis to assess meat quality changes.

Response: The goal of our work was not to monitor the sensory quality of deer meat, because it was contaminated with pathogenic bacteria. Sensory analysis of meat using the method in this study is not possible. Our study is focused on inhibiting the growth of L. monocytogenes and other microorganisms using lower temperatures.

Comments 5: Extend the storage period to evaluate long-term microbial stability and shelf life.

Response: We think that 14 days for sus vide meat is maximum considering the microbiological quality of the samples with temperature used. Maximal number of sous vide meat is 30 day but for temperature 65-95°C.

Comments 6: Incorporate additional parameters (pH, water activity) to monitor meat quality.

Response: It was done.

Comments 7: Explore the mechanisms of action of PNEO against target microorganisms.

Response: The inhibitory mechanisms of PNEO against L. monocytogenes encompassed perturbation of cellular morphology, elevation in reactive oxygen species levels, augmentation of lipid oxidation extent, hyperpolarization of membrane potential, and reduction in intracellular ATP concentration.  The antimicrobial effect of PNEO is mainly due to their constituent phenolic compounds. The mode of action of PNEO may include damage to cytoplasmic membranes, protein denaturation, cytoplasmic coagulation, and depletion of the proton motive force of microorganisms.

Comments 8: In the methods part: How do the temperature and time settings used in this study compared to those in other research on meat preservation?

Response: In the material and methods section, we used temperature and time that were already published in other studies. it should be taken into account that the goal of the work was inactivation or effect of PNEO on the growth and reproduction of L. monocytogenes in a meat model. Similar times and temperature are also useful in other studies, e.g. on cheese, vegetables and other types of meat

Gál, R., Čmiková, N., Kačániová, M., & Mokrejš, P. (2023). Salvia officinalis Essential Oil as an Antimicrobial Agent against Salmonella enterica in Sous-Vide Beef During Storage. https://doi.org/10.20944/preprints202310.0754.v1

Kačániová, M., Čmiková, N., Kluz, M. I., Akacha, B. B., Saad, R. B., Mnif, W., Waszkiewicz-Robak, B., Garzoli, S., & Hsouna, A. B. (2024). Anti-Salmonella Activity of Thymus serpyllum Essential Oil in Sous Vide Cook–Chill Rabbit Meat. Foods, 13(2), 200. https://doi.org/10.3390/foods13020200

Gál, R., Čmiková, N., Prokopová, A., & Kačániová, M. (2023). Antilisterial and Antimicrobial Effect of Salvia officinalis Essential Oil in Beef Sous-Vide Meat during Storage. Foods, 12(11), 2201. https://doi.org/10.3390/foods12112201

Hernández, H., Fraňková, A., Klouček, P., & Banout, J. (2018). The Effect of the Application of Thyme Essential Oil on Microbial Load During Meat Drying. Journal of Visualized Experiments, 133. https://doi.org/10.3791/57054-

Comments 9: In the results part: 1- Quantify the reduction in microbial counts achieved in this study compared to other studies using sous-vide and essential oils or other preservation methods. 2- Compare the shelf-life extension observed in this study to that reported in other studies using similar or different preservation techniques. 3- Analyze the economic feasibility of the method compared to other preservation techniques, considering factors such as energy consumption, labor costs, and material expenses.

Response: Yes, quantity the reduction in microbial counts achieved in this study is comparable with other study with same conditions.  The economic feasibility is low, the price of plant essential oils is low, and only 1% concentration is used in our methodology, also the costs of sample packaging are not high and are acceptable, as well as laboratory analyzes are normally used for any analysis in a microbiology laboratory. For example, when packaged in a modified atmosphere, the costs of used gases during packaging are high, so this method can be used in undemanding economic conditions.

Gál, R., Čmiková, N., Kačániová, M., & Mokrejš, P. (2023). Salvia officinalis Essential Oil as an Antimicrobial Agent against Salmonella enterica in Sous-Vide Beef During Storage. https://doi.org/10.20944/preprints202310.0754.v1

Kačániová, M., Čmiková, N., Kluz, M. I., Akacha, B. B., Saad, R. B., Mnif, W., Waszkiewicz-Robak, B., Garzoli, S., & Hsouna, A. B. (2024). Anti-Salmonella Activity of Thymus serpyllum Essential Oil in Sous Vide Cook–Chill Rabbit Meat. Foods, 13(2), 200. https://doi.org/10.3390/foods13020200

Gál, R., Čmiková, N., Prokopová, A., & Kačániová, M. (2023). Antilisterial and Antimicrobial Effect of Salvia officinalis Essential Oil in Beef Sous-Vide Meat during Storage. Foods, 12(11), 2201. https://doi.org/10.3390/foods12112201

Hernández, H., Fraňková, A., Klouček, P., & Banout, J. (2018). The Effect of the Application of Thyme Essential Oil on Microbial Load During Meat Drying. Journal of Visualized Experiments, 133. https://doi.org/10.3791/57054-

Comments 8: All tables and figures need more attention to introduce the main results. It is not clear.

Response: All tables and graphs are described in the text. Due to the huge number of identified samples, it is not possible to express the results in any other way as shown in the tables. A supplementary file is attached to the article, where the results are expressed graphically. If necessary, we can exchange images for tables. The identification of species is expressed by crown images that are commonly used in the identification of microorganisms by various methods and are obtained automatically through the used program.

By addressing these points, the authors can significantly strengthen the study and increase its potential impact.

Overall, major revisions are recommended before considering the manuscript for publication.

Round 2

Reviewer 1 Report (Previous Reviewer 1)

Comments and Suggestions for Authors

The authors of the manuscript “Characterization and Antimicrobial Power of Piper Nigrum Essential Oil: Application for Extending Shelf Life of Sous-Vide Deer Meat” have made the necessary revisions to the manuscript based on the reviewers' comments. In addition, they have addressed each of the reviewers' questions regarding the research. The manuscript, in its current state, is ready for publication on the journal's platform.

Reviewer 3 Report (New Reviewer)

Comments and Suggestions for Authors

The authors significantly revised the manuscript, answered to all comments and questions. I think that the article has improved significantly and can be accepted for publication.

This manuscript is a resubmission of an earlier submission. The following is a list of the peer review reports and author responses from that submission.

Round 1

Reviewer 1 Report

Comments and Suggestions for Authors

The manuscript “Characterization and Antimicrobial Power of Piper Nigrum Essential Oil: Application for Extending Shelf Life of Sous-Vide Deer Meat” shows significant advancements in food preservation using sous-vide technology in combination with essential oils. However, the authors need to clarify the following points:

  1. What would be the optimal concentration of PNEO to maximize the reduction of Listeria monocytogenes without compromising the organoleptic properties of red deer meat?
  2. How does the treatment with PNEO and the sous-vide method affect the sensory properties (taste, texture, aroma) of red deer meat?
  3. How does the effectiveness of the treatment with PNEO and sous-vide compare to other preservation methods, such as irradiation or the use of other essential oils?
  4. What is the effect of different combinations of time and temperature in the sous-vide method on the reduction of microbial load and the quality of the meat?
  5. What is the specific mechanism of action of PNEO on Listeria monocytogenes and other microorganisms present in the meat?
  6. Is the treatment with PNEO and sous-vide equally effective against other common foodborne pathogens, such as Salmonella or Escherichia coli?
  7. Are there significant interactions between the compounds in PNEO and the biochemical components of the meat that could influence the safety or quality of the final product?
Comments on the Quality of English Language

It is recommended that the manuscript be reviewed by a native English speaker, as it contains numerous grammatical and formatting errors.

Author Response

Reviewer #1

The Authors are very grateful to the Reviewer for their valuable comments. We want to thank the Reviewer for the time devoted to point out constructive and important comments to improve our paper. Based on your questions listed in the review, we present some points in the discussion.

The manuscript “Characterization and Antimicrobial Power of Piper Nigrum Essential Oil: Application for Extending Shelf Life of Sous-Vide Deer Meat” shows significant advancements in food preservation using sous-vide technology in combination with essential oils. However, the authors need to clarify the following points:

Comment 1: What would be the optimal concentration of PNEO to maximize the reduction of Listeria monocytogenes without compromising the organoleptic properties of red deer meat?

Response:

In general, the use of EOs as food ingredients has several limitations. Their intense aroma may alter the

organoleptic properties of foods, which may be unacceptable for consumers. With this drawback in mind, food processors have to use EOs at low concentrations, which, in turn, due to the possible interaction with food constituents (fats, starch or proteins) and external factors (light, oxidation or heating), may result in poor antimicrobial activity or even inefficiency. Therefore, researchers are looking for alternative ways of use in order to achieve both consumer acceptance and food safety. An effective strategy is to apply EOs in the active packaging of food products. This involves the migration of active compounds from the packaging to the food, providing protection against microorganisms. The appropriate concentration of PNEO has been proposed based on data previously published [1-3] on other types of meat and our own results. The optimal concentration for the application of EOs is between 0.5 and 2%. However, to broaden our research and optimize the experimental part, we are currently working with other EOs and their applications on different types of meat.

1-Pesavento, G.; Calonico, C.; Bilia, A.R.; Barnabei, M.; Calesini, F.; Addona, R.; Mencarelli, L.; Carmagnini, L.; Di Martino, M.C.; Lo Nostro, A. Antibacterial Activity of Oregano, Rosmarinus and Thymus Essential Oils against Staphylococcus aureus and Listeria monocytogenes in Beef Meatballs. Food Control 201554, 188–199.

2-Gouveia, A.R.; Alves, M.; Silva, J.A.; Saraiva, C. The Antimicrobial Effect of Rosemary and Thyme Essential Oils against Listeria monocytogenes in Sous Vide Cook-Chill Beef During Storage. Procedia Food Sci. 20167, 173–176.

3-Khaleque, M.A.; Keya, C.A.; Hasan, K.N.; Hoque, M.M.; Inatsu, Y.; Bari, M.L. Use of Cloves and Cinnamon Essential Oil to Inactivate Listeria monocytogenes in Ground Beef at Freezing and Refrigeration Temperatures. LWT 201674, 219–223.

Comment 2: How does the treatment with PNEO and the sous-vide method affect the sensory properties (taste, texture, aroma) of red deer meat?

Response: The goal of our work was not to monitor the sensory quality of deer meat, because it was contaminated with pathogenic bacteria.

Comment 3: How does the effectiveness of the treatment with PNEO and sous-vide compare to other preservation methods, such as irradiation or the use of other essential oils?

Response: Improper heat treatment is the main problem in sous-vide processes applied at low temperatures. Therefore, sous-vide cooking has recently been combined with the use of natural antioxidants such as EOs to improve the efficiency of the cooking process in terms of food safety during the preservation phase. This is in accordance with the latest trends in the food industry where different processing procedures could be combined to promote process efficiency. In addition, there is an increased awareness towards the replacement of synthetic antioxidants with natural ones, in food processing; therefore, sous-vide cooking supplemented by the addition of natural antioxidants could be a prospective tool for food preservation. Several studies have shown that this concept is a promising alternative in preserving the quality of meat products. On this topic, there are several studies where EOs based on spices such as oregano, thyme in combination with the sous vide technique, have been used on fish or on pork, beef and poultry to inactivate bacterial growth. For this reason, we consider our work innovative, because the present study is the first to have investigated the effects of the use of PNEO on venison.

Comment 4: What is the effect of different combinations of time and temperature in the sous-vide method on the reduction of microbial load and the quality of the meat?

Response: Due to the low and precise temperatures used and the minimal impact on the organoleptic and nutritional properties of food, SV technology is gaining popularity. It is well known that sensory and flavor properties are important criteria for consumer perception and acceptance of the organoleptic attributes of food and the physicochemical changes that occur in food during processing. Although the culinary aspect of SV has been well established, scientific research on the microbiological safety aspect has yet to be fully considered and remains a concern. Evaluating this aspect of the technology through literature review and analysis will provide foundations and insights that will guide further research. Currently, in contrast to its previous position as a sophisticated technique for individual caterers, SV is increasingly being accepted by the mass production sector as a food processing method. The main benefit of applying SV technology is the optimal preservation of quality without altering the organoleptic properties of the food. Compared to traditional cooking methods, the precise temperature control employed in the technique offers more choices in cooking and texture. In addition, the use of heat-stable vacuum bags improves shelf life and can enhance taste and nutritional value. However, the low cooking temperatures used in this technique could lead to microbiological concerns related to food safety due to non-compliance with pasteurization standards/processes. As a result, the survival of damaged foodborne pathogenic microorganisms after treatment could pose a threat to food consumers or lead to microbial-mediated product degradation and shortening of shelf life. Consequently, the food industry could adapt this technique to meet this need. In SV, foods undergo minimal heat treatment, maintaining their freshness, flavor, texture and even color, while maintaining their microbial safety. However, it is important to fully understand the microbiological implications of this technique in terms of controlling pathogenic organisms in foods and extending the shelf life of foods. Research on shelf-life extension effects has shown that microbial growth is reduced after treatment. Therefore, a synergistic effect has been proposed such as, SV combined with high-pressure processing (HPP) may prove more effective in extending the shelf-life of food products.

Comment 5: What is the specific mechanism of action of PNEO on Listeria monocytogenes and other microorganisms present in the meat?

Response: The antimicrobial effect of EOs is mainly due to their constituent phenolic compounds. The mode of action of EOs may include damage to cytoplasmic membranes, protein denaturation, cytoplasmic coagulation, and depletion of the proton motive force. The inhibitory effect of EOs on L. monocytogenes has been documented in numerous studies. The obtained results indicated that LM populations increased during seven and fourteen days of storage at 4 ◦C in the control groups but decreased when exposed to EOs treatment. At concentrations of 0.5% and 1%, EOs limited the growth of LM in meat at both temperatures, with better effects at the higher dose of 1%. In conclusion, EOs slowed the growth rates of L. monocytogenes populations compared to control during 14 days of storage at 4 ◦C. Further data show the efficacy of EO (1% v/w) in a meat model against two levels of an L. monocytogenes cocktail (3 and 6 log CFU/g) combined with storage at 4 ◦C for 14 days.

Comment 6: Is the treatment with PNEO and sous-vide equally effective against other common foodborne pathogens, such as Salmonella or Escherichia coli?

Response: PNEO and other EOs inhibit the growth of Salmonella and Escherichia coli. Our study builds on our results and those obtained in previous works using different meat models and different Eos [1-4]. This is why we decided to combine PNEO from L. monocytogenes and we are already working with venison treated with pepper-based essential oils to evaluate their inhibitory capacity on other microorganisms.

1-Gál, R., Čmiková, N., Kačániová, M., & Mokrejš, P. (2023). Salvia officinalis Essential Oil as an Antimicrobial Agent against Salmonella enterica in Sous-Vide Beef During Storage. https://doi.org/10.20944/preprints202310.0754.v1

2-Kačániová, M., Čmiková, N., Kluz, M. I., Akacha, B. B., Saad, R. B., Mnif, W., Waszkiewicz-Robak, B., Garzoli, S., & Hsouna, A. B. (2024). Anti-Salmonella Activity of Thymus serpyllum Essential Oil in Sous Vide Cook–Chill Rabbit Meat. Foods, 13(2), 200. https://doi.org/10.3390/foods13020200

3-Gál, R., Čmiková, N., Prokopová, A., & Kačániová, M. (2023). Antilisterial and Antimicrobial Effect of Salvia officinalis Essential Oil in Beef Sous-Vide Meat during Storage. Foods, 12(11), 2201. https://doi.org/10.3390/foods12112201

4-Hernández, H., Fraňková, A., Klouček, P., & Banout, J. (2018). The Effect of the Application of Thyme Essential Oil on Microbial Load During Meat Drying. Journal of Visualized Experiments, 133. https://doi.org/10.3791/57054-v

Comment 7: Are there significant interactions between the compounds in PNEO and the biochemical components of the meat that could influence the safety or quality of the final product?

Response: Black pepper (Piper nigrum L.) is one of the main flavoring agents in meat processing. With a relatively high percentage of terpenoids (limonene, α- and β-pinene and caryophyllene), EOs isolated from black pepper (BPEOs), show a strong antioxidant effect, as well as a preservative effect against a broad spectrum of microorganisms. BPEO was added as a natural preservative in fresh pork loin at concentrations of 0 to 0.5%. All batches were stored at 4 °C for 9 days. The study showed that BPEO delayed lipid oxidation and reduced the growth of Enterobacteriaceae and Pseudomonas spp. in fresh pork. In another study, the effect of BPEO coating (0.05 and 0.1%) on lipid oxidation and sensory quality (aroma) of ham was examined. The authors suggested that the use of BPEO has a strong potential to suppress lipid oxidation and improve the sensory acceptability of ham during long-term storage (4 months at room temperature). Overall, the results suggest that BPEO could be used as a natural antioxidant in meat products.

Comment 8: Comments on the Quality of English Language

It is recommended that the manuscript be reviewed by a native English speaker, as it contains numerous grammatical and formatting errors.

Response: English language was corrected.

Reviewer 2 Report

Comments and Suggestions for Authors

Attach file

Comments on the Quality of English Language

Author Response

Reviewer #2

The manuscript ID molecules 3092249 is entitled: “Characterization and Antimicrobial Power of Piper nigrum Essential Oil: Application for Extending Shelf Life of Sous-Vide Deer Meat”. The authors described the efficiency of vacuum sealing in conjunction with Piper nigrum essential oil treatment to preserve red deer meat samples for sous-vide cooking and to assess the suitability of minimal meat processing. The effectiveness of these methods was evaluated by analyzing quality indicators and quantifying Listeria monocytogenes in red deer meat samples inoculated with the bacteria during refrigerated storage. Tables are in good quality and with adequate information for justified the results of identified components and biological assay. In conclusion: Investigation revealed a diverse microbiome in sous-vide cooked red deer meat through mass spectrometry analysis. In this study was identified other common species like Kocuria salsicia, Pseudomonas taetrolens, and Pseudomonas fragi. The use of essential oil at a concentration of 1% had a positive impact on reducing L. monocytogenes. These findings offer valuable insights for food producers, suggesting that sous-vide cooking with black pepper essential oil can naturally inhibit L. monocytogenes growth, making game meat safer for storage at appropriate temperatures. The manuscript is suitable for publication in Molecules.

The Authors are very grateful to the Reviewer for their valuable comments. We want to thank the Reviewer for the time devoted to point out constructive and important comments to improve our paper.

Minor corrections:

Comment 1: p.2 line 69: … Approximately 55 to 60 % of the animal's weight comprises … change by… Approximately 55 to 60% of the animal's weight comprises …. Check throughout the manuscript

Response: Done

Comment 2: p.12 line 441: … sesquiterpene (E)-caryophyllene (25.9%), which was followed … change by … sesquiterpene 441 (E)-caryophyllene (25.9%), which was followed …. Check throughout the manuscript

Response: Done

Comment 3: p.18 line 551, reference 4: … Deer (Cervus Elaphus) or Wild Boar (Sus Scrofa) Meat.… change by … Deer (Cervus elaphus) or Wild Boar (Sus scrofa) Meat …. Check scientific names in italic throughout all references.

Response: Done

Reviewer 3 Report

Comments and Suggestions for Authors

Comments to the Authors

The manuscript entitled “Characterization and Antimicrobial Power of Piper Nigrum Essential Oil: Application for Extending Shelf Life of Sous-Vide Deer Meat”. Although the topic may be very interesting, I suggest to reject the manuscript.

In my opinion, there is no basis for critical evaluation of the data on the plant material used to obtain the essential oil (botanical identification and time of harvest, as an example). In addition, this is already pulverized material of which no process data can be reported. 

Author Response

Reviewer #3

The Authors are very grateful to the Reviewer for their valuable comments. We want to thank the Reviewer for the time devoted to point out constructive and important comments to improve our paper.

Comments to the Authors

Comment 1: The manuscript entitled “Characterization and Antimicrobial Power of Piper Nigrum Essential Oil: Application for Extending Shelf Life of Sous-Vide Deer Meat”. Although the topic may be very interesting, I suggest to reject the manuscript.

Response: In your reviewer comments is it not mentioned the reason.

Comment 2: In my opinion, there is no basis for critical evaluation of the data on the plant material used to obtain the essential oil (botanical identification and time of harvest, as an example). In addition, this is already pulverized material of which no process data can be reported. 

Response Botanical identification has been corrected. In light of the comment made in this subsection, we must defend ourselves, because in the section materials and methods it is clearly stated that the chemical composition and preparation of the essential oil have already been published in another work, and for this reason these data have not been mentioned in the present article. The aim of the work was: The research focused on examining the efficiency of vacuum sealing in combination with treatment with Piper nigrum essential oil (PNEO) to preserve red deer meat samples for sous-vide cooking and to evaluate the suitability of minimal meat processing. The effectiveness of these methods was assessed by analyzing quality indicators and evaluating the bacterial growth of Listeria monocytogenes in red deer meat samples inoculated with the bacteria during refrigerated storage.